Earth System
**Science**
**Data**

Open Access | Discussions

# National CO$_2$ budgets (2015–2020) inferred from atmospheric CO$_2$ observations in support of the Global Stocktake

Brendan Byrne[1], David F. Baker[2], Sourish Basu[3,4], Michael Bertolacci[5], Kevin W. Bowman[1,6], Dustin Carroll[7,1], Abhishek Chatterjee[1], Frédéric Chevallier[8], Philippe Ciais[8], Noel Cressie[5,1], David Crisp[1], Sean Crowell[9], Feng Deng[10], Zhu Deng[11], Nicholas M. Deutscher[12], Manvendra K. Dubey[13], Sha Feng[14], Omaira E. García[15], David W. T. Griffith[12], Benedikt Herkommer[16], Lei Hu[17,18], Andrew R. Jacobson[17,18], Rajesh Janardanan[19], Sujong Jeong[20], Matthew S. Johnson[21], Dylan B. A. Jones[10], Rigel Kivi[22], Junjie Liu[1,23], Zhiqiang Liu[24], Shamil Maksyutov[19], John B. Miller[17], Scot M. Miller[25], Isamu Morino[19], Justus Notholt[26], Tomohiro Oda[27,28], Christopher W. O'Dell[2], Young-Suk Oh[29], Hirofumi Ohyama[19], Prabir K. Patra[30], Hélène Peiro[9], Christof Petri[26], Sajeev Philip[31], David F. Pollard[32], Benjamin Poulter[3], Marine Remaud[8], Andrew Schuh[2], Mahesh K. Sha[33], Kei Shiomi[34], Kimberly Strong[10], Colm Sweeney[17], Yao Té[35], Hanqin Tian[36,37], Voltaire A. Velazco[12,38], Mihalis Vrekoussis[39,26], Thorsten Warneke[26], John R. Worden[1], Debra Wunch[10], Yuanzhi Yao[36], Jeongmin Yun[20], Andrew Zammit-Mangion[5], and Ning Zeng[28,4]

[1]Jet Propulsion Laboratory, California Institute of Technology, Pasadena, CA, USA
[2]Cooperative Institute for Research in the Atmosphere, Colorado State University, Fort Collins, CO, USA
[3]NASA Goddard Space Flight Center, Global Modeling and Assimilation Office, Greenbelt, MD, USA
[4]Earth System Science Interdisciplinary Center, College Park, MD, USA
[5]School of Mathematics and Applied Statistics, University of Wollongong, Australia
[6]Joint Institute for Regional Earth System Science and Engineering, University of California, Los Angeles, CA, USA
[7]Moss Landing Marine Laboratories, San José State University, Moss Landing, CA, USA
[8]Laboratoire des Sciences du Climat et de L'Environnement, LSCE/IPSL, CEA-CNRS-UVSQ, Université Paris-Saclay, 91191 Gif-sur-Yvette, France
[9]University of Oklahoma, Norman, OK, USA
[10]Department of Physics, University of Toronto, Toronto, Ontario, Canada
[11]Department of Earth System Science, Tsinghua University, Beijing, China
[12]Centre for Atmospheric Chemistry, School of Earth, Atmospheric and Life Sciences, University of Wollongong, Wollongong, NSW, Australia
[13]Earth System Observation, Los Alamos National Laboratory, Los Alamos, NM, USA
[14]Atmospheric Sciences and Global Change Division, Pacific Northwest National Laboratory, Richland, WA, USA
[15]Izaña Atmospheric Research Center (IARC), State Meteorological Agency of Spain (AEMet), Tenerife, Spain
[16]Institut for Meteorology and Climate Research (IMK-ASF), Karlsruhe Institute of Technology (KIT), Karlsruhe, Germany
[17]NOAA Global Monitoring Laboratory, Boulder, CO, USA
[18]Cooperative Institute for Research in Environmental Sciences, University of Colorado Boulder, Boulder, CO, USA
[19]Satellite Observation Center, Earth System Division, National Institute for Environmental Studies, Tsukuba, Japan
[20]Department of Environmental Planning, Graduate School of Environmental Studies, Seoul National University, Seoul, Republic of Korea
[21]NASA Ames Research Center, Moffett Field, CA, USA
[22]Space and Earth Observation Centre, Finnish Meteorological Institute, Sodankylä, Finland
[23]Division of Geological and Planetary Sciences, California Institute of Technology, Pasadena, CA, USA
[24]Laboratory of Numerical Modeling for Atmospheric Sciences & Geophysical Fluid Dynamics, Institute of Atmospheric Physics, Chinese Academy of Sciences, Beijing, China





[25]Department of Environmental Health and Engineering, Johns Hopkins University, Baltimore, MD 21218, United States of America

[26]Institute of Environmental Physics, University of Bremen, Bremen, Germany

[27]Earth from Space Institute, Universities Space Research Association, Columbia, MD, USA

[28]Department of Atmospheric and Oceanic Science, University of Maryland, USA

[29]Global Atmosphere Watch Team, Climate Research Department, National Institute of Meteorological Sciences, Republic of Korea

[30]Research Institute for Global Change, Japan Agency for Marine-Earth Science and Technology (JAMSTEC), Yokohama, 236-0001, Japan

[31]Centre for Atmospheric Sciences, Indian Institute of Technology Delhi, New Delhi, India

[32]National Institute of Water & Atmospheric Research Ltd (NIWA), Lauder, New Zealand

[33]Royal Belgian Institute for Space Aeronomy (BIRA-IASB), Brussels, Belgium

[34]Japan Aerospace Exploration Agency (JAXA), Tsukuba, Japan

[35]Laboratoire d'Etudes du Rayonnement et de la Matière en Astrophysique et Atmosphères (LERMA-IPSL), Sorbonne Université, CNRS, Observatoire de Paris, PSL Université, 75005 Paris, France.

[36]International Center for Climate and Global Change Research, College of Forestry, Wildlife and Environment, Auburn University, Auburn, AL 36849, USA

[37]Schiller Institute for Integrated Science and Society, and Department of Earth and Environmental Sciences, Boston College, Chestnut Hill, MA 02467, USA

[38]Deutscher Wetterdienst (DWD), Hohenpeissenberg, Germany.

[39]Climate and Atmosphere Research Center (CARE-C), The Cyprus Institute, Nicosia, Cyprus

**Correspondence:** Brendan Byrne (brendan.k.byrne@jpl.nasa.gov)

**Abstract.** Accurate accounting of emissions and removals of $CO_2$ is critical for the planning and verification of emission reduction targets in support of the Paris Agreement. Here, we present a pilot dataset of country-specific net carbon exchange (NCE; fossil plus terrestrial ecosystem fluxes) and terrestrial carbon stock changes aimed at informing countries' carbon budgets. These estimates are based on "top-down" NCE outputs from the v10 Orbiting Carbon Observatory (OCO-2) modeling inter-

5    comparison project (MIP), wherein an ensemble of inverse modeling groups conducted standardized experiments assimilating OCO-2 column-averaged dry-air mole fraction ($X_{CO_2}$) retrievals (ACOS v10), in situ $CO_2$ measurements, or combinations of these data. The v10 OCO-2 MIP NCE estimates are combined with "bottom-up" estimates of fossil fuel emissions and lateral carbon fluxes to estimate changes in terrestrial carbon stocks, which are impacted by anthropogenic and natural drivers. These flux and stock change estimates are reported annually (2015–2020) as both a global $1° \times 1°$ gridded dataset and as a country-

10    level dataset. Across the v10 OCO-2 MIP experiments, we obtain increases in the ensemble median terrestrial carbon stocks of 3.29–4.58 $PgCO_2 \, yr^{-1}$ (0.90–1.25 $PgC \, yr^{-1}$). This is a result of broad increases in terrestrial carbon stocks across the northern extratropics, while the tropics generally have stock losses but with considerable regional variability and differences between v10 OCO-2 MIP experiments. We discuss the state of the science for tracking emissions and removals using top-down methods, including current limitations and future developments towards top-down monitoring and verification systems.



# 1 Introduction

To reduce the risks and impacts of climate change, the Paris Agreement aims to limit the global average temperature increase to well below 2 °C above pre-industrial levels and pursue efforts to limit these increases to less than 1.5 °C. To this end, each Party to the Paris Agreement agreed to prepare and communicate successive Nationally Determined Contributions (NDCs) of greenhouse gas (GHG) emission reductions. Collective progress toward this goal of the Paris Agreement is evaluated in Global Stocktakes (GSTs), which are conducted at five-year intervals; the first GST is scheduled in 2023. The outcome of each GST is then used as input, or as a "ratchet mechanism", for new NDCs that are meant to encourage greater ambition.

In support of the first GST, Parties to the Paris Agreement are compiling inventories of GHG emissions and removals to inform their progress toward the emission-reduction targets in their individual NDCs. These inventories are generally estimated using "bottom-up" approaches, wherein $CO_2$ emission estimates are based on activity data and emission factors while $CO_2$ removals by sinks are based on inventories of carbon stock changes and models. This approach allows for explicit characterization of $CO_2$ emissions and removals into the five main sectors specified in the 2006 IPCC Guidelines for National Greenhouse Gas Inventories (IPCC, 2006): Energy, Industrial Processes and Product Use (IPPU), Agriculture, Forestry and Other Land Use (AFOLU), Waste, and Other. Bottom-up methods can provide precise and accurate country-level emission estimates when the activity data and emission factors are well quantified and understood, such as for the fossil fuel combustion category of the energy sector in many countries. However, these estimates can also have considerable uncertainty when the emission processes are challenging to quantify (such as for AFOLU) or if the activity data are inaccurate or missing. In addition, these estimates do not capture carbon emissions and removals from unmanaged systems, which are not directly considered in the Paris Agreement, but nevertheless impact the global carbon budget and growth rate of atmospheric $CO_2$.

As a complement to these accounting-based inventory efforts, an independent "top-down" assessment of net surface–atmosphere $CO_2$ fluxes may be obtained from ground-based, airborne and space-based observations of atmospheric $CO_2$ mole fractions. These top-down methods have undergone rapid improvements in recent years, as recognized in the 2019 Refinement to the 2006 IPCC Guidelines for National Greenhouse Gas Inventories (IPCC, 2019). And, although these methods were not deemed to be a standard tool for verification of conventional inventories, a number of countries (UK, Switzerland, USA, and New Zealand) have adopted atmospheric inverse modeling as a verification system in national inventory reports. Initially, these countries have focused on non-$CO_2$ gasses (e.g., EPA, 2022), but top-down assessments of the $CO_2$ budget are now underdevelopment in New Zealand (https://niwa.co.nz/climate/research-projects/carbon-watch-nz). Furthermore, significant investments towards building anthropogenic $CO_2$-emissions monitoring and verification support capacity are ongoing within the European Commission's Copernicus Program (see Sect. 8.2.1).

In top-down $CO_2$ flux estimation, the net surface–atmosphere $CO_2$ fluxes are inferred from atmospheric $CO_2$ observations using state-of-the-art atmospheric $CO_2$ inversion systems (e.g., Peiro et al., 2022). This approach provides spatially- and temporally-resolved estimates of surface–atmosphere fluxes for land and ocean regions from which country-level annual land–atmosphere $CO_2$ fluxes can be estimated. The impact of fossil fuel (and usually fire $CO_2$ emissions) on the observations are accounted for in the inversions by prescribing maps of those emissions and assuming that they are perfectly known. Thus,





fossil fuel and fire $CO_2$ emissions are not diagnosed yet by these inversions, but net surface–atmosphere $CO_2$ fluxes from the terrestrial biosphere and oceans are. Terrestrial carbon stock changes can then be calculated by combining net surface–atmosphere $CO_2$ fluxes with estimates of fossil fuel emissions and horizontal ("lateral") fluxes occurring within the terrestrial biosphere or between the land and ocean (Kondo et al., 2020). One example of a lateral flux is harvested agricultural products, where carbon is sequestered from the atmosphere by photosynthesis in one region but then this carbon is harvested and exported

to another region as agricultural products. Similarly, carbon sequestered by photosynthesis in a forest can be leached away by streams and rivers, and then exported to the ocean. These lateral carbon fluxes are not directly identifiable in atmospheric $CO_2$ measurements, but accounting for their impact is required in order to convert net land fluxes into stock changes. These estimated terrestrial carbon stock changes reflect the combined impact of direct anthropogenic activities and changes to both managed and unmanaged ecosystems in response to rising $CO_2$, climate change, and disturbance events (such as fires).

The top-down budgets presented here extend several previous studies that have developed approaches to compare inversion results to United Nation Framework Convention of Climate Change (UNFCCC) inventories. Ciais et al. (2021) proposed a protocol for reporting bottom-up and top-down fluxes so that they can be compared consistently. Chevallier (2021) noted that inversion results for terrestrial $CO_2$ fluxes should be restricted to managed lands and applied a managed land mask to the gridded fluxes of the CAMS $CO_2$ inversions for the comparison to UNFCCC values in ten large countries or groups of

countries. Deng et al. (2022) compared $CO_2$, $CH_4$ and $N_2O$ fluxes from inversion ensembles available from the Global Carbon Project. For $CO_2$, they used six $CO_2$ flux estimates from inverse models that assimilated measurements from the global air-sample network, filtered their results over managed lands and corrected them for $CO_2$ fluxes induced by lateral processes to compare with carbon stock changes reported to the UNFCCC by a set of 12 countries. We expand upon these previous studies by providing top-down $CO_2$ budgets from the v10 Orbiting Carbon Observatory Model Intercomparison Project (v10 OCO-2

MIP), wherein an ensemble of inverse modeling groups conducted standardized experiments assimilating OCO-2 column-averaged dry-air mole fraction ($X_{CO_2}$) retrievals (retrieved with version 10 of the Atmospheric $CO_2$ Observations from Space (ACOS) full-physics retrieval algorithm), in situ $CO_2$ measurements, or combinations of these data. This allows us to quantify the sensitivity of top-down carbon budget estimates to the inversion modeling system and the atmospheric $CO_2$ dataset used to constrain flux estimates.

This paper is outlined as follows. The remainder of Sect. 1 describes the objectives of this work (Sect. 1.1) and provides background information on both the global carbon cycle (Sect. 1.2) and top-down atmospheric $CO_2$ inversions (Sect. 1.3). Section 2 defines the carbon cycle fluxes of interest. Section 3 describes the flux datasets and their uncertainties, including: fossil fuel emissions, the v10 OCO-2 MIP, riverine fluxes, wood fluxes, crop fluxes, and the net terrestrial carbon stock loss. Section 4 provides an evaluation of the v10 OCO-2 MIP flux estimates. Section 5 presents two metrics for interpreting the

top-down constraints on the $CO_2$ budget. Section 6 gives a description of the dataset, Sect. 7 shows the characteristics of the dataset, and Sect. 8 discusses current limitations and future directions. Finally, Sect. 9 gives the conclusions of this study.





## 1.1 Objectives

This is a pilot project designed to start a dialogue between the top-down research community, inventory compilers, and the GHG assessment community to identify ways that top-down $CO_2$ flux estimates can help inform country-level carbon budgets (see Worden et al. (2022) for a similar pilot methane dataset). To meet this objective, the primary goal of this work is to provide two products: (1) annual net surface–atmosphere $CO_2$ fluxes and (2) annual changes in terrestrial carbon stocks. These products are provided annually over the six-year period 2015–2020 on both a $1° \times 1°$ global grid and as country-level totals with error characterization.

These products are intended to be used to help inform inventory development and identify areas for future research in both top-down and bottom-up approaches. Including, informing strategies for operational top-down carbon cycle products that can be used for tracking combined changes in managed and unmanaged carbon stocks and help quantify the impact of emission reduction activities.

## 1.2 Overview of the carbon cycle

The carbon cycle describes the movement of carbon between various reservoirs in the Earth system. Carbon cycles between the atmosphere, oceans, and terrestrial biosphere through a number of pathways, while the burning of fossil fuels and cement production release geologic carbon to the atmosphere ($40.0 \pm 3.3 \, \mathrm{PgCO_2 \, yr^{-1}}$ or $10.9 \pm 0.9 \, \mathrm{PgC \, yr^{-1}}$ over 2010–2019; Canadell et al., 2021). On an annual net basis, roughly half of the emitted $CO_2$ from anthropogenic sources is absorbed by terrestrial ecosystems and oceans (Friedlingstein et al., 2022), reducing the rate of atmospheric $CO_2$ increase ($18.7 \pm 0.08 \, \mathrm{PgCO_2 \, yr^{-1}}$ or $5.1 \pm 0.02 \, \mathrm{PgC \, yr^{-1}}$ over 2010–2019; Canadell et al., 2021). Here we briefly review the movement of carbon between the reservoirs, and how these processes are modulated by human activities.

Fluxes of carbon between the atmosphere and ocean are driven by the difference in partial pressures of $CO_2$ between seawater and air, resulting in roughly balancing fluxes from the ocean-to-atmosphere and atmosphere-to-ocean of $\sim 293 \, \mathrm{PgCO_2 \, yr^{-1}}$ ($\sim 80 \, \mathrm{PgC \, yr^{-1}}$) each way (Ciais et al., 2013), with a residual net atmosphere-to-ocean flux due to increasing atmospheric $CO_2$ ($9.2 \pm 2.2 \, \mathrm{PgCO_2 \, yr^{-1}}$ or $2.5 \pm 0.6 \, \mathrm{PgC \, yr^{-1}}$ over 2010–2019; Canadell et al., 2021). Regional variations in the solubility and saturation of $CO_2$ in ocean waters drive net fluxes, with net fluxes to the atmosphere in upwelling regions, such as the eastern boundary of basins and in equatorial zones (McKinley et al., 2017). Meanwhile, there is net removals by the ocean in western boundary currents and at extratropical latitudes (McKinley et al., 2017). Within the oceans, circulation patterns, mixing, and biologic activity act to redistribute carbon.

On land, terrestrial ecosystems remove atmospheric carbon through photosynthesis, referred to as Gross Primary Production (GPP) (Fig. 1). GPP draws roughly $440 \, \mathrm{PgCO_2 \, yr^{-1}}$ ($120 \, \mathrm{PgC \, yr^{-1}}$) from the atmosphere (Anav et al., 2015). Roughly half of this carbon is emitted back to the atmosphere by plants through autotrophic respiration, while the remaining carbon is used to generate plant biomass and is referred to as Net Primary Production (NPP). On an annual basis, the carbon sequestered through NPP is roughly balanced by carbon loss through a number of processes. The largest of these processes is heterotrophic respiration, which is the respiratory emission of $CO_2$ (from the dead organic matter and soil carbon pools) by heterotrophic



organisms, and accounts for 82–95% of NPP (Randerson et al., 2002). The combination of heterotrophic and authotrophic respiration is called ecosystem respiration ($R_{eco}$). The remaining processes have smaller magnitudes, but are still critical for determining the carbon balance of ecosystems. Biomass burning, the emission of carbon to the atmosphere through combustion, releases roughly 7.3 $PgCO_2$ $yr^{-1}$ (2 PgC $yr^{-1}$) to the atmosphere on an annual basis, but with considerable interannual variability (van der Werf et al., 2017). Carbon can also be emitted from the terrestrial biosphere to the atmosphere in the form

of carbon monoxide (CO), methane ($CH_4$) and other biologic volatile organic compounds (BVOCs), which are oxidized to $CO_2$ in the atmosphere. Rivers move carbon in the form of dissolved inorganic carbon (DIC), dissolved organic carbon (DOC), and particulate organic carbon (POC). This carbon of terrestrial origin is partly transported to the open ocean, partly released to the atmosphere from inland waters and estuaries, and partly buried in aquatic or marine sediments. Finally, anthropogenic activities such as harvesting of crop and wood products result in lateral transport of carbon, such that the removal of atmospheric $CO_2$

through NPP and emission of atmospheric $CO_2$ through respiration (e.g., decomposition in a landfill) or combustion (e.g., burning of biofuels) occurs in different regions. See Fig. 1 for an illustration of these fluxes.

Globally, there is a long-term net uptake of atmospheric $CO_2$ by the land (approximately -6.6 $PgCO_2$ $yr^{-1}$ or -1.8 $PgC_2$ $yr^{-1}$ over 2010–2019; Canadell et al., 2021), which is the residual of an emission due to net land use change (5.9 ± 2.6 $PgCO_2$ $yr^{-1}$ or 1.6 ± 0.7 $PgC$ $yr^{-1}$ over 2010–2019; Canadell et al., 2021) and removal by other terrestrial ecosystems (12.6 ± 3.3 $PgCO_2$ $yr^{-1}$

or 3.4 ± 0.9 $PgC$ $yr^{-1}$ over 2010–2019; Canadell et al., 2021). This removal is partially driven by direct feedbacks between increasing $CO_2$ and the biosphere, such as $CO_2$ fertilization of photosynthesis and increased water use efficiency. Carbon-climate feedbacks also lead to both increases and decreases in terrestrial carbon stocks: for example, warming at high latitudes leads to a more productive biosphere but it also leads to increased plant and soil respiration (Kaushik et al., 2020; Walker et al., 2021; Canadell et al., 2021; Crisp et al., 2022). In addition, the release of nitrogen through anthropogenic energy and fertilizer

use may also drive increased carbon sequestration by the terrestrial biosphere. Regrowth of forests in previously cleared areas, especially in the extratropics, is also thought to be an important uptake term (Kondo et al., 2018; Cook-Patton et al., 2020). Currently, the relative impact of each of these contributions to long-term terrestrial carbon sequestration is poorly known, and likely varies between biomes and climates.

While the existence of a long-term global land sink is supported through a number of lines of evidence (Ballantyne et al.,

2012; Keeling and Graven, 2021), regional-scale emissions and removals are less well quantified. Regional-scale carbon sequestration can differ substantially from the global mean and can be impacted by the regional climate, disturbance events (Frank et al., 2015; Wang et al., 2021), and anthropogenic activities (Caspersen et al., 2000; Harris et al., 2012). The need to better quantify regional-scale emissions and removals of carbon has motivated much of the recent expansion of in situ $CO_2$ observing networks, the launch of space-based $CO_2$ observing systems, and the development of $CO_2$ inversion systems.

## 1.3 Background on atmospheric $CO_2$ inversions

Atmospheric $CO_2$ inversions estimate the underlying net surface–atmosphere $CO_2$ fluxes from atmospheric $CO_2$ observations, and this is what is meant by the "top-down" approach (Bolin and Keeling, 1963; Tans et al., 1990; Enting et al., 1995; Gurney et al., 2002; Peiro et al., 2022). In this approach, an atmospheric chemical transport model (CTM) is employed to relate



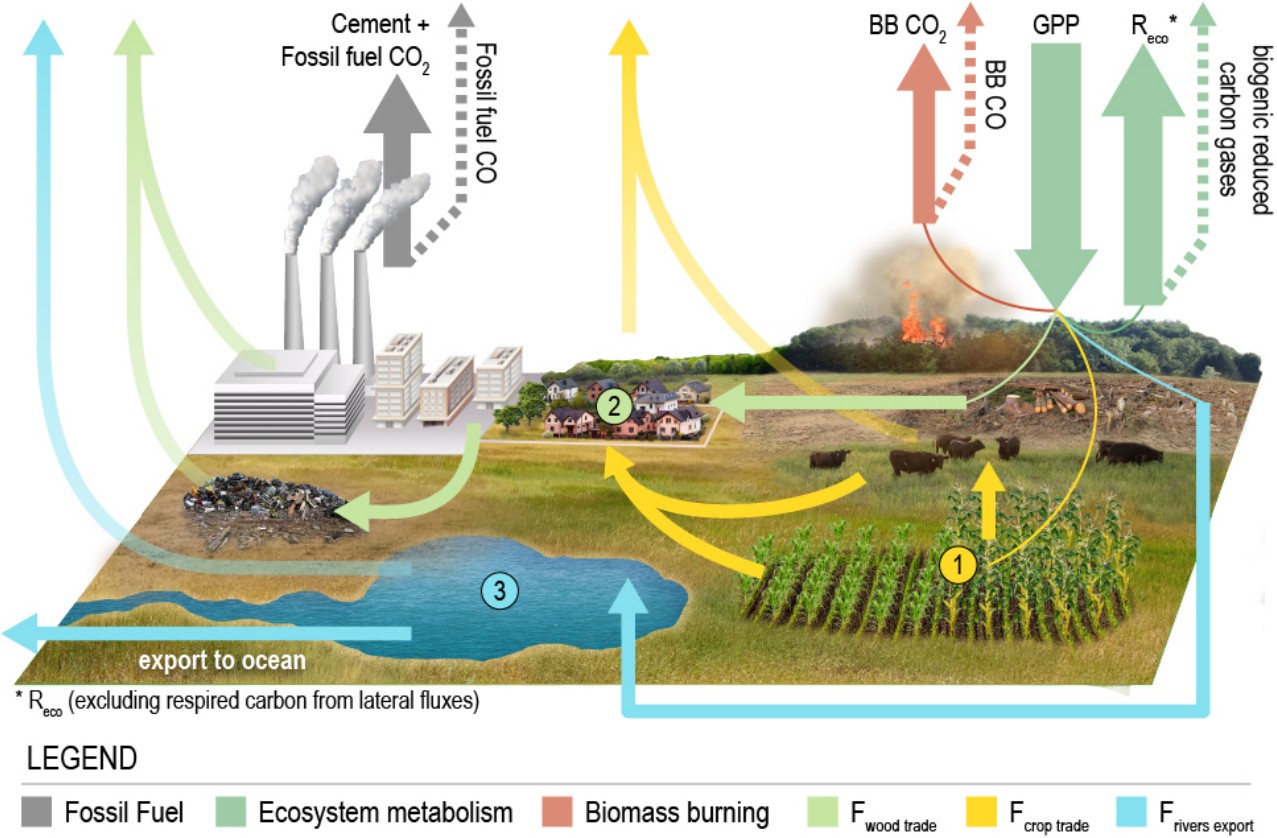

**Figure 1.** $CO_2$ is removed from the atmosphere through photosynthesis (GPP) and then emitted back to the atmosphere through a number of processes. Three processes move carbon laterally on Earth's surface, such that emissions of $CO_2$ occur in a different region than removals: (1) Agriculture; harvested crops are transported to urban areas and to livestock, which are themselves exported to urban areas. $CO_2$ is respired to the atmosphere in livestock or urban areas. (2) Forestry; logged carbon is transported to urban and industrial areas, then emitted to through decomposition in a landfill or combustion as a biofuel. (3) Water cycle; carbon is leached from soils into water bodies, such as lakes. The carbon is then either deposited, released to the atmosphere, or transported to the ocean (Regnier et al., 2022). Arrows show carbon fluxes and colors indicate whether the flux is associated with (grey) fossil fuel emissions, (dark green) ecosystem metabolism, (red) biomass burning, (light green) forestry, (yellow) agriculture, or (blue) the water cycle. Semi-transparent arrows show fluxes that move between the surface and atmosphere, while solid arrows show fluxes that move between land regions. Dashed arrows show surface–atmosphere fluxes of reduced carbon species that are oxidized to $CO_2$ in the atmosphere. For simplicity, a cement carbonation sink, volcano emissions, and a weathering sink are not included in this figure.

surface–atmosphere $CO_2$ fluxes to observed atmospheric $CO_2$ mole fractions. As an inverse problem, the upwind $CO_2$ fluxes are estimated from the downwind observed $CO_2$ mole fractions. The surface $CO_2$ fluxes are adjusted so that forward-simulated $CO_2$ mole fractions better match the $CO_2$ measurements while considering the uncertainty statistics on the observations, transport, and prior surface fluxes.




The atmospheric $CO_2$ inversion problem is generally ill-posed, such that the solution is underdetermined by the observational constraints. In this case, additional information is required to produce a unique solution and prevent overfitting of the data
(Lawson and Hanson, 1974; Tarantola, 2005). Typically, this is performed using Bayesian inference, where prior mean fluxes and their uncertainties provide additional information required to estimate fluxes (Rayner et al., 2019). Prior mean fluxes of net ecosystem exchange are usually obtained from dynamic global vegetation models, while prior mean air-sea fluxes are derived from surface water partial pressure of $CO_2$ ($pCO_2$) datasets or from ocean models (e.g., Peiro et al., 2022). The resulting posterior flux estimates combine the constraints on surface fluxes from atmospheric $CO_2$ data with the prior knowledge of the
fluxes. If there is a high density of assimilated $CO_2$ observations, then the posterior fluxes will be more strongly impacted by the assimilated data, whereas, in regions with sparse observational coverage, the posterior fluxes will generally remain similar to the prior fluxes (assuming similar prior flux uncertainties across regions).

Measurements of atmospheric $CO_2$ best inform diffuse biosphere–atmosphere fluxes on large spatial scales. This is because $CO_2$ has a long atmospheric lifetime, such that the perturbation to atmospheric $CO_2$ due to emissions and removals from
individual processes and locations gets mixed in the atmosphere (Gloor et al., 2001; Liu et al., 2015). For example, the measurements of $CO_2$ at Mauna Loa, Hawaii, provide a good estimate of the global-scale changes of $CO_2$ surface fluxes. Inferring smaller-scale flux signals requires a high density of $CO_2$ observations (to capture gradients in atmospheric $CO_2$) and accurate modeling of atmospheric transport (to relate the measurements with surface fluxes). The accuracy of flux estimates depend on a number of factors, particularly the accuracy and precision of the data, transport model, and prior constraints. Stringent require-
ments on the accuracy of space-based column-averaged dry-air mole fraction ($X_{CO_2}$) retrievals are required to infer surface fluxes (Chevallier et al., 2005a; Miller et al., 2007). Biases in $X_{CO_2}$ retrievals from the Orbiting Carbon Observatory (OCO-2) related to spectroscopic errors, solar zenith angle, surface properties, and atmospheric scattering by clouds and aerosols have been identified (Wunch et al., 2017b). However, intensive research has reduced retrieval errors over time (O'Dell et al., 2018; Kiel et al., 2019). As will be shown in Sect. 4.1, biases in OCO-2 $X_{CO_2}$ retrievals over land are thought to be relatively
small, although regionally structured biases may be present. However, OCO-2 $X_{CO_2}$ retrievals over oceans may contain more large-scale spatially coherent retrieval errors that can adversely impact flux estimates.

Accurate atmospheric transport is critical for correctly relating surface–atmosphere fluxes to observations. Due to computational constraints, CTMs are typically run offline with coarsened meteorological fields relative to the parent Numerical Weather Prediction model, which has been shown to introduce systematic transport errors in some configurations (Yu et al.,
2018; Stanevich et al., 2020). In addition, these offline CTMs have been shown to have large-scale systematic differences in transport associated with the implementation of transport algorithms (Schuh et al., 2019, 2022). These errors appear to be of the same order as the retrieval biases, although the patterns in time and space are different. Systematic errors related to model transport (and errors in prior information) can partially be accounted for by performing multiple inversions that differ in CTM and prior constraints employed. This motivates inversion model intercomparison projects (MIPs), such as the OCO-2
MIP project (see Sect. 3.2; Crowell et al., 2019; Peiro et al., 2022). From these ensembles of inversions, estimates of both systematic errors (accuracy) and random errors (precision) can be obtained from the model spread.



## 2 Definitions

In this work, we focus on the carbon budget of Earth's land area, including aquatic systems such as rivers and lakes. In particular, we consider fluxes of carbon between the land and the atmosphere, and lateral carbon transport processes on land and between the land and ocean. We define the following annual net carbon fluxes (see Fig. 2 for an schematic representation of these fluxes):

- **Fossil fuel and cement emissions** (FF): The burning of fossil fuels and release of carbon due to cement production, representing a flux of carbon from the land surface (geologic reservoir) to the atmosphere.

- **Net Biosphere exchange** (NBE): Net flux of carbon from the terrestrial biosphere to the atmosphere due to biomass burning (BB) and ecosystem respiration ($R_{eco}$) minus Gross Primary Production (GPP) (i.e., $NBE = BB + R_{eco} - GPP$). It includes both anthropogenic processes (e.g., deforestation, reforestation, farming) and natural processes (e.g., climate-variability-induced carbon fluxes, disturbances, recovery from disturbances).

- **Terrestrial Net Carbon Exchange** (NCE): Net flux of carbon from the surface to the atmosphere. For land, NCE can be defined as:

$$NCE = NBE + FF \tag{1}$$

- **Lateral crop flux** ($F_{crop\,trade}$): The lateral flux of carbon in (positive) or out (negative) of a region due to agriculture.

- **Lateral wood flux** ($F_{wood\,trade}$): The lateral flux of carbon in (positive) or out (negative) of a region due to wood product harvesting and usage.

- **Lateral river flux** ($F_{rivers\,export}$): The lateral flux of carbon in (positive) or out (negative) of a region transported by the water cycle.

- **Net terrestrial carbon stock loss** ($\Delta C_{loss}$): Positive values indicate a loss (decrease) of terrestrial carbon stocks (organic matter stored on land), including above- and below-ground biomass in ecosystems and biomass contained in anthropogenic products (lumber, cattle, etc). This is calculated as:

$$\Delta C_{loss} = NBE - F_{crop\,trade} - F_{wood\,trade} - F_{rivers\,export} \tag{2}$$

- **Net terrestrial carbon stock gain** ($\Delta C_{gain}$): Positive values indicate a gain (increase) of terrestrial carbon stocks, and is the negative of $\Delta C_{loss}$:

$$\Delta C_{gain} = -\Delta C_{loss} \tag{3}$$

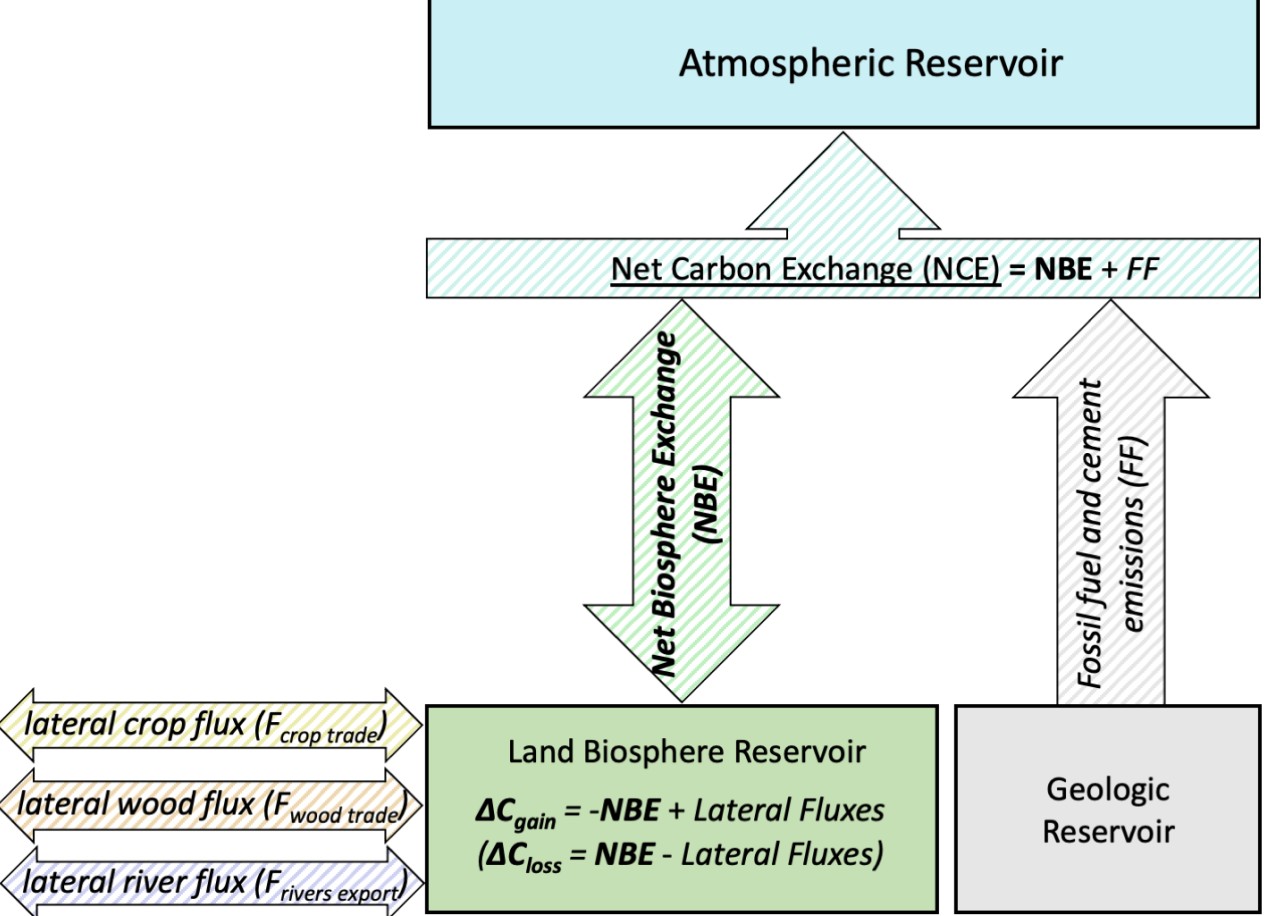

**Figure 2.** Carbon fluxes for a given land region, such as a country. Boxes with solid backgrounds show reservoirs of carbon. Arrows with hatched shading show fluxes between reservoirs. NCE is underlined to emphasize that this quantity is estimated from the atmospheric $CO_2$ measurements using top-down methods. Italicized quantities are obtained from bottom-up datasets ($FF$, $F_{\text{crop trade}}$, $F_{\text{wood trade}}$, $F_{\text{rivers export}}$). Bold quantities are derived in this study from the top-down and bottom-up datasets (**NBE**, $\Delta\mathbf{C}_{\text{gain}}$, $\Delta\mathbf{C}_{\text{loss}}$).

## 2.1 Country and regional aggregation

To aggregate gridded $1° \times 1°$ flux estimates to country totals we use a country mask (Center for International Earth Science Information Network - CIESIN - Columbia University, 2018). We also provide NCE and $\Delta C_{\text{loss}}$ estimates for several country groupings. A number of regional intergovernmental organizations are included: the Association of Southeast Asian Nations (ASEAN), the African Union (AU) and each of its sub-regions (North, South, West, East, and Central), the Community of Latin American and Caribbean States plus Brazil (CELAC+Brazil), the Economic Cooperation Organization (ECO), the European Union (EU), and the South Asian Association for Regional Cooperation (SAARC). We also include some geographic regions,





specifically North America, the Middle East and Europe. Countries included in these groupings are listed in the supplementary materials (Text S1).

## 3 Flux datasets

Here, we describe the methodologies and datasets for estimating FF (Sect. 3.1), NCE (Sect. 3.2), country-level $F_{\text{rivers export}}$ (Sect. 3.3), country-level $F_{\text{crop trade}}$ and country-level $F_{\text{wood trade}}$ (Sect. 3.4). Gridded lateral fluxes are estimated using a
somewhat different approach, and are described in Sect. 3.5. Finally, we describe how these data are used to estimate $\Delta C_{\text{loss}}$ (Sect. 3.6).

### 3.1 Fossil fuel and cement emissions

Gridded $1° \times 1°$ fossil $CO_2$ emissions, including from cement production, are calculated as follows. Monthly gridded emissions up to 2019 are taken from the 2020 version of the Open-source Data Inventory for Anthropogenic $CO_2$ (ODIAC2020, 2000–
2019) emission data product (Oda and Maksyutov, 2011; Oda et al., 2018). The 2020 emissions were not part of ODIAC, but were projected using the Carbon Monitor (CM) emission data product (https://carbonmonitor.org/, downloaded 19th May 2021). For each month in 2020 and later, the ratio between that month's emissions and the emissions from the same month in 2019 was calculated from the CM emission data. Since CM provides daily emissions per sector for a handful of major emitting countries and the globe, CM emissions are summed over sectors and days in each month to create monthly total emissions per
named country and the rest of the world (RoW). The ratio of each (post-2019) month's emission to the same month in 2019 is then calculated per named country and RoW, then distributed over a $1° \times 1°$ grid assuming homogeneity of the ratio over each named country and RoW. 2019 ODIAC emissions for that month are then multiplied by the ratio to generate $1° \times 1°$ monthly emissions after 2019. While this method loses the information of day-to-day variability provided by CM, this is a conscious choice to be consistent over the entire inversion period. Finally, we impose day-of-week and hour-of-day variations on these
fluxes following the Temporal Improvements for Modeling Emissions by Scaling (TIMES) diurnal and day-of-week scaling (Nassar et al., 2013). The $1° \times 1°$ uncertainty map is based on the combination of the global level FF uncertainty (one-sigma of 4.2%, Andres et al., 2014) and the grid level emission differences due to the different disaggregation methods (Oda et al., 2015). Note that these FF uncertainties are not considered in the inversions used for this product development.

    Country-level fossil fuel emission estimates are obtained by aggregating the $1° \times 1°$ estimates using the country mask.
Uncertainties on country-level estimates are calculated using the fractional uncertainties of Andres et al. (2014).

### 3.2 Net Carbon Exchange (NCE) and Net Biosphere Exchange (NBE)

We employ results from the v10 OCO-2 MIP, which is an international collaboration of atmospheric $CO_2$ inversion modelers that produces ensembles of $CO_2$ surface–atmosphere flux estimates by assimilating space-based OCO-2 retrievals of $X_{CO_2}$ and in situ $CO_2$ measurements. The v10 OCO-2 MIP is updated from the v9 OCO-2 MIP described in Peiro et al. (2022). Updates
to the v10 OCO-2 MIP are presented here with additional details available at https://gml.noaa.gov/ccgg/OCO$_2$_v10mip/.

**Table 1.** Inversions specifications for each v10 OCO-2 MIP ensemble member.

| Simulation name (reference) | Transport model | Driving meteorology | Meteorology resolution (degrees) | Prior NEE | Prior air-sea | Prior fire | Inverse Method |
|---|---|---|---|---|---|---|---|
| AMES (Philip et al., 2019, 2022) | GEOS-Chem | MERRA-2 | 4° × 5° | CASA-GFED4.1s | CT2019OI | GFEDv4.1s | 4D-Var |
| Baker (Baker et al., 2006b, 2010) | PCTM | MERRA-2 | 1° × 1.25° prior, 4° × 5° opt | CASA-GFED3 | Landschützer v4.4 | GFEDv4 | 4D-Var |
| CAMS (Chevallier et al., 2005b) | LMDz | ERA5 | 1.9° × 3.75° | ORCHIDEE (climatological) | CMEMS | GFED4.1s | Variational |
| CMS-Flux^a (Liu et al., 2021a) | GEOS-Chem | MERRA-2 | 4° × 5° | CARDAMOM | MOM-6 | CARDAMOM | 4D-Var |
| COLA (Liu et al., 2021b) | GEOS-Chem | MERRA-2 | 4° × 5° | VEGAS | Rödenbeck 2021 | VEGAS | EnKF |
| CSU | GEOS-Chem | MERRA-2 | 4° × 5° | SiB-4 w/ MERRA-2 | Landschützer v18 | GFED4.1s | synthesis |
| CT (Jacobson et al., 2020) | TM5 | ERA5 | 2° × 3°/ 1° × 1° | CT2019 CASA GFED4.1s | CT2019OI | CT2019 CASA GFED4.1s | EnKF |
| JHU^a (Miller et al., 2020) (Chen et al., 2021b, a) | GEOS-Chem | MERRA-2 | 4° × 5° | CASA-GFED 4.1s | Takahashi | GFED4.1s | geostatistical/ 4D-Var |
| LoFI^b (Weir et al., 2021) | GEOS GCM | GEOS CGM | 0.5° × 0.625° | CASA-GFED3 | LoFI Takahasi | QFED | N/A |
| NIES (Maksyutov et al., 2021) | NIES-TM/ FLEXPART | ERA-5/ JRA-55 | 3.75° × 3.75°/ 0.1° × 0.1° | Zeng 2020 | Landschützer 2020 | GFAS | 4D-Var |
| OU | TM5 | ERA-Interim | 4° × 6° | CASA-GFED3 | Takahashi | GFEDv3 | 4D-Var |
| TM5-4DVar | TM5 | ERA-Interim | 2° × 3° | SiB-CASA | CT2019 Opt Clim | GFEDv4 | 4D-Var |
| UT (Deng et al., 2014, 2016) | GEOS-Chem | GEOS-FP | 4° × 5° | BEPS | Takahashi | GFEDv4 | 4D-Var |
| WOMBAT (Zammit-Mangion et al., 2022) | GEOS-Chem | MERRA-2 | 2° × 2.5° | SiB-4 w/ MERRA-2 | Landschutzer 2020 | GFED4.1s | Synthesis with MCMC |

[a] Due to time constraints, this ensemble member is not included in this product; [b]This is not a traditional inversion and is not included in this product.





The v10 OCO-2 MIP consists of a number of inversion systems that perform a set of experiments following a standard protocol. Here, we include fluxes from 11 of the 14 MIP models (Table 1; CMS-Flux and JHU were excluded due to time constraints and LoFI was excluded because it employs a non-traditional inversion approach that does not follow the MIP protocol). There are five v10 OCO-2 MIP experiments that each ensemble member performs, which differ by the data that is

assimilated ($CO_2$ datasets described in Sect. 3.2.1):

- **IS**: assimilates in situ $CO_2$ mole fraction measurements from an international observational network,

- **LNLG**: ACOS v10 land nadir and land glint total column dry-air mole fractions ($X_{CO_2}$) from OCO-2,

- **LNLGIS**: assimilates both in situ and ACOS v10 OCO-2 land nadir and glint $X_{CO_2}$ retrievals together,

- **OG**: assimilates ACOS v10 OCO-2 ocean glint $X_{CO_2}$ retrievals

- **LNLGOGIS**: assimilates all the above datasets together.

For each experiment, each inversion group imposes a common fossil fuel emission dataset identical to the one described in Sect. 3.1. All other prior flux estimates were chosen independently by each modeling group and are listed in Table 1. The inversions assimilate the standardized v10 OCO-2 and in situ data from 6 September 2014 through 31 March 2021 (see Sect 3.2.1), with the length of spin-up period and in situ data assimilated during that period being left up to the discre-

tion of each group in the MIP. Each modeling group submitted net air–sea fluxes and NBE across 2015–2020, interpolated from the native resolution to a $1° \times 1°$ spatial grid at monthly resolution, which are publicly available for download from https://gml.noaa.gov/ccgg/OCO$_2$_v10mip/.

The performance of each atmospheric $CO_2$ inversion was evaluated through comparisons of the posterior $CO_2$ mole-fraction field (i.e., $CO_2$ fields simulated forward with the posterior fluxes) against independent in situ $CO_2$ measurements and OCO-2

$X_{CO_2}$ retrievals that were withheld from the assimilation for validation, as well as $X_{CO_2}$ retrievals from the Total Column Carbon Observing Network (TCCON; Wunch et al., 2011). The evaluation of the experiments is presented in Sect. 4, with additional analysis available from the v10 OCO-2 MIP website.

For this study, the best estimate of NCE is taken to be the ensemble median for each experiment (denoted $\overline{NCE_{experiment}}$). The uncertainty in NCE is calculated as an estimate of the standard deviation (denoted $\sigma_{NCE}$) using the interquartile range

(IQR) of the flux-inversion ensemble:

$$\sigma_{NCE} = \frac{IQR(NCE)}{1.35}. \tag{4}$$

For country-level fluxes, the NCE estimates are first aggregated to country totals for each ensemble member before calculating the median and standard deviation. This is done because there are spatial covariances between $1° \times 1°$ grid cells. Thus, first aggregating regions for each ensemble member accurately propagates the aggregate differences between regions across the

ensemble members.



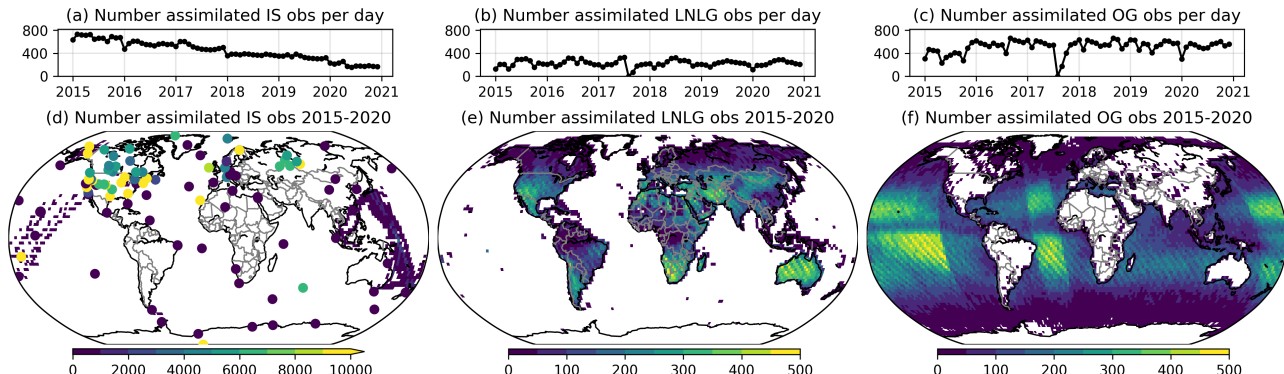

**Figure 3.** Assimilated observations for IS and LNLG v10 MIP experiments. Number of monthly (a) in situ $CO_2$ measurements and (b) ACOS v10 OCO-2 land nadir and land glint $X_{CO_2}$ retrievals binned into 10 s averages, and (c) ACOS v10 OCO-2 ocean glint $X_{CO_2}$ retrievals binned into 10 s averages. Spatial distribution of (d) in situ (e) ACOS v10 OCO-2 land $X_{CO_2}$ retrievals, and (f) ACOS v10 OCO-2 ocean $X_{CO_2}$ retrievals over 2015–2020. Shipboard and aircraft in situ $CO_2$ measurements are aggregated to a $2° \times 2°$ spatial grid, surface site measurements are shown as scattered points, and ACOS v10 OCO-2 $X_{CO_2}$ retrievals are shown aggregated to a $2° \times 2°$ spatial grid.

The NBE estimate is calculated by subtracting the ODIAC Fossil Fuel emissions from NCE. The variance in NBE is then taken to be the sum of the variances of NCE and FF:

$$\sigma_{\text{NBE}}^2 = \sigma_{\text{NCE}}^2 + \sigma_{\text{FF}}^2 \tag{5}$$

### 3.2.1 Atmospheric $CO_2$ data included in v10 OCO-2 MIP

In situ $CO_2$ measurements (Fig. 3a,d) are drawn from five data collections made available in Obspack format (Masarie et al., 2014). Those source ObsPacks and their references are listed in Table 2. The majority of data are from the openly available GLOBALVIEW+ program, but with some additional provisional data for 2020–21, and data from other programs not participating in the GLOBALVIEW+ project. $CO_2$ measurements are broadly divided into two categories: those measurements we identify as suitable for assimilation, and other measurements not suitable for assimilation.

In $CO_2$ inverse analyses, uncertainties ascribed to in situ measurements are a combination of the uncertainty in the measurement and a representativeness error from the forward model inability to accurately simulate the measurement (due to aspects like a coarse model grid). To characterize the representativeness error, we used an empirical scheme based on simulations from the v7 OCO-2 MIP (Crowell et al., 2019). In situ $CO_2$ measurements are simulated in a forward simulation, and then the model-data mismatch statistics are calculated to characterize the representativeness errors at each measurement location and 295 for each season. Although this was the standard method for characterizing uncertainties for modeled in situ measurements, each v10 OCO-2 MIP group was free to choose how to set the uncertainties in their specific set-ups.

Of the in situ measurements designated as being appropriate for assimilation, about 5% were withheld for cross-validation purposes. These data were chosen to be as independent as possible from the measurements that were assimilated. For quasi-continuous measurements, such as those taken every 15 minutes at NOAA tall towers, measurements were withheld for entire

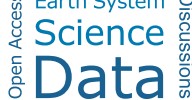

**Table 2.** In situ $CO_2$ measurement collections used in the v10 OCO-2 MIP, with the total number of measurements between 6 September 2014 and 1 January 2021, and the numbers of measurements assimilated and withheld for cross-validation in the same period. More than 95% of the in situ measurements come from the GLOBALVIEW+ and $CO_2$ NRT ObsPacks, both of which are publicly available at https://gml.noaa.gov/ccgg/obspack/data.php.

| ObsPack name | Total No. Measurements | Assimilated | Withheld | Reference |
|---|---|---|---|---|
| obspack_CO$_2$_1_GLOBALVIEWplus_v6.1_2021-03-01 | 9,611,095 | 766,179 | 38,483 | Schuldt et al. (2021b) |
| obspack_CO$_2$_1_NRT_v6.1.1_2021-05-17 | 755,477 | 62,011 | 2,996 | Schuldt et al. (2021a) |
| obspack_CO$_2$_1_NIES_Shipboard_v3.0_2020-11-10 | 418,496 | 216,963 | 12,766 | Tohjima et al. (2005); Nara et al. (2017) |
| obspack_CO$_2$_1_AirCore_v4.0_2020-12-28 | 55,620 | | | Baier et al. (2021) |
| obspack_multi-species_1_manaus_profiles_v1.0_2021-05-20 | 3,194 | | | Miler et al. (2021) |
| TOTAL | 10,843,882 | 1,045,153 | 54,245 | |





days: we chose 5% of the days in the dataset, and we withheld every assimilable measurement on that day. This is also how $CO_2$ measurements on NIES ships were treated. Entire aircraft profiles in the NOAA light-aircraft profiling network are assumed to consist of vertically correlated measurements, so entire profiles were withheld: we chose 5% of aircraft profiles to withhold. Most flask sites have measurement sampling protocols intended to ensure independence; they are often taken at weekly or biweekly intervals during meteorological conditions meant to allow regional background air masses to be sampled. Thus, we

chose to withhold 5% of assimilable flask measurements. We also verified that datasets at the same site were withheld on the same days; aircraft profiles over tower sites were, for instance, withheld on the same days that tower data were withheld.

OCO-2 land (Fig. 3b,e) and ocean (Fig. 3c,f) $X_{CO_2}$ retrievals are performed using version 10 of NASA's Atmospheric $CO_2$ Observations from Space (ACOS) full-physics retrieval algorithm (O'Dell et al., 2018). A common set of OCO-2 retrieval "super-obs" data were derived from these retrievals and were assimilated by each modeling group. These "super-obs" are

obtained by aggregating retrievals into 10 s averages (which better match the coarse transport models grid cells used in the inversions) following the same procedure as the v9 OCO-2 MIP (Peiro et al., 2022). Specifically, individual scenes within the 10 s span are weighted according to the inverse of the square of the $X_{CO_2}$ uncertainty (standard deviations) produced by the retrieval, and correlations of +0.3 for land scenes and +0.6 for ocean scenes are assumed when calculating the uncertainty on the 10-second averages (see Sect. 3.2.1 of Baker et al., 2022); transport model errors are also considered (based on Schuh

et al., 2019). Only 10 s spans with 10 or more good quality retrievals were used (sparser data being thought to be more prone to cloud-related biases). In the same vein as was done for the in situ data, $X_{CO_2}$ data from 5% of the orbits (entire orbits were withheld), chosen at random, were withheld for evaluation purposes.

### 3.3 Country-level $F_{\text{rivers export}}$

Rivers transport carbon laterally across land regions (e.g., to a lake) and from the land to the ocean. This lateral transport must

be accounted for to quantify the total change in terrestrial carbon in a given region. However, there is considerable uncertainty in lateral carbon flux by rivers. To account for this, we use two independent estimates of country-level totals: one from the Dynamic Land Ecosystem Model (DLEM), and the other based on Deng et al. (2022) who use the Global NEWS model (Mayorga et al., 2010) and observations across COastal Segmentation and related CATchments (COSCATs) (Meybeck et al., 2006) that include DIC (of atmospheric origin), DOC and POC. These datasets cover 2015–2019. For 2020, we impose the

2015–2019 mean.

The DLEM is a process-based terrestrial ecosystem model that couples biophysical, soil biogeochemical, plant physiological and riverine processes with vegetation and land-use dynamics to simulate and predict the vertical fluxes, lateral fluxes, and storage of water, carbon, GHGs, and nutrient dynamics in terrestrial ecosystems and their interfaces with the atmosphere and land-ocean continuum (Tian et al., 2010, 2015a). There are three major processes involved in simulating the export of water,

carbon, and nutrients from land surface to the coastal ocean: 1) the generation of runoff and leachates, 2) the leaching of water, carbon and nutrients from land to river networks in the form of overland flow and base flow, and 3) transport of riverine materials along river channels from upstream areas to coastal regions. The key processes and parameterization in the DLEM have been described in previous publications regarding the water discharge (Liu et al., 2013; Tao et al., 2014), riverine carbon



fluxes (Ren et al., 2015, 2016; Tian et al., 2015b; Yao et al., 2021), and riverine nitrogen fluxes (Yang et al., 2015; Tian et al.,
2020) from the terrestrial ecosystem to coastal oceans. The newly improved DLEM aquatic module better addresses processes
within global small streams, which were recognized as hotpots of GHG emissions (Yao et al., 2020, 2021). DLEM produces
estimates of the land loadings of carbon species (DIC, DOC, and POC), $CO_2$ degassing and carbon burial during transporting,
and the exports of carbon (DIC, DOC, and POC) to the ocean for 105 basin-level segmentations (modified from COSCATs)
(Meybeck et al., 2006). To estimate country totals, we map the basin carbon loss across land by assuming that the net carbon
flux occurs uniformly across each basin. We then use the country mask to estimate the country totals for each region.

Deng et al. (2022) estimate the lateral carbon export by rivers to the coast minus the imports from rivers entering in each
country (for relevant cases), including DOC, POC and DIC of atmospheric origin. Estimates of DOC, POC and DIC are
obtained from the Global NEWS model (Mayorga et al., 2010), with a correction based on Resplandy et al. (2018) so that the
global total exported to the coastal ocean is $2.86 \, \mathrm{PgCO_2 \, yr^{-1}}$ ($0.78 \, \mathrm{PgC \, yr^{-1}}$). Deng et al. (2022) perform a correction to the
Global NEWS estimates to remove the contribution of lithogenic carbon, using the methodology of Ciais et al. (2021).

For the analysis that follows, we estimate country-level totals of riverine lateral carbon fluxes by combining the estimates of
DLEM with those of Deng et al. (2022). We take the mean of the two estimates to be the best estimate and take the magnitude
of the difference between the estimates to be the one-sigma uncertainty. Figure S1 shows the 2015–2019 mean annual net
riverine lateral carbon fluxes. Fluxes are uniformly negative, implying a net flux of carbon from the land to the ocean and
reduction in stored carbon for all countries. Fluxes are most negative in tropical rainforest and tropical monsoon climates, and
they are smallest in more arid regions.

### 3.4 Country-level $F_{\mathrm{wood \, trade}}$ and $F_{\mathrm{crop \, trade}}$

Wood and crop products are traded between nations. We estimate the annual lateral fluxes of carbon due to this trade following
the approaches of Deng et al. (2022) and Ciais et al. (2021). This approach utilizes crop and wood trade data compiled by
the Food and Agriculture Organization of the United Nations (FAO, http://www.fao.org/faostat/en/#data). The crop flux was
estimated from the annual trade balance of 171 crop commodities calculated for each country. For wood products, we use the
bookkeeping model of Mason Earles et al. (2012) to calculate the fraction of imported carbon in wood products that is oxidized
in each of 270 countries during subsequent years. One-sigma uncertainties in country-level fluxes are assumed to be 30% of
the mean value. This dataset covers 2015–2019. For 2020, we assume fluxes equal to the 2015–2019 mean. The net crop and
wood lateral fluxes and their uncertainties are shown in Fig. S2.

### 3.5 $1° \times 1°$ lateral flux estimates

Lateral fluxes at a higher resolution ($1° \times 1°$) follow similar principles to national values but were estimated separately with
different implementation choices. High-resolution proxy data (satellite-derived NPP, population or livestock maps, etc.) enabled
subnational disaggregation. For each $1° \times 1°$ grid cell, we assume the standard deviation of the mean flux to be 30% for
$F_{\mathrm{wood \, trade}}$ and $F_{\mathrm{crop \, trade}}$, and 60% for $F_{\mathrm{rivers \, export}}$.





## 3.6  Estimate of carbon stock loss ($\Delta C_{\mathrm{loss}}$)

Finally, we calculate $\Delta C_{\mathrm{loss}}$ using Eqn. 2 with the datasets described above. Assuming that the components contributing to $\Delta C_{\mathrm{loss}}$ are independent, we calculate the uncertainty on $\Delta C_{\mathrm{loss}}$ by combining the uncertainties (one standard deviations) from the component fluxes in quadrature:

$$\sigma^2_{\Delta C_{\mathrm{loss}}} = \sigma^2_{\mathrm{NBE}} + \sigma^2_{F_{\mathrm{crop\ trade}}} + \sigma^2_{F_{\mathrm{wood\ trade}}} + \sigma^2_{F_{\mathrm{rivers\ export}}} \tag{6}$$

## 4  Evaluation of v10 OCO-2 MIP experiments

The performance of top-down $CO_2$ flux estimates can be impacted by a number of factors, including biases in the assimilated data, model transport, prior constraints, and in the inversion architectures. Therefore, evaluating the performance of v10 OCO-2 MIP fluxes against independent datasets is critical for assuring high quality flux estimates. Here, we evaluate the v10 OCO-2 MIP experiments in two ways. First, we compare the posterior $CO_2$ fields against independent $CO_2$ measurements (Sect. 4.1). Second, we compare the inferred air–sea $CO_2$ flux against estimates based on surface ocean $CO_2$ partial pressure (pCO$_2$) measurements (Sect. 4.2).

### 4.1  Evaluation of posterior $CO_2$ fields

We consider four atmospheric $CO_2$ datasets:

1. Withheld in situ $CO_2$ measurements. These are measurements contained in the Obspack collection described in Sect. 3.2.1 but intentionally withheld for evaluation purposes. Independence from the assimilated data is ensured following the steps described in Sect. 3.2.1.

2. $X_{CO_2}$ retrievals from the TCCON. These data are acquired from a network of ground-based Fourier Transform Spectrometers measuring direct solar spectra from which $X_{CO_2}$ is retrieved (Wunch et al., 2011). For this analysis, we include 30 TCCON sites listed in table A1. These data are filtered and aggregated following the method outlined in Appendix C of Crowell et al. (2019).

3. Withheld OCO-2 land glint and land nadir $X_{CO_2}$ retrievals. These data could have been assimilated, but they are intentionally withheld for evaluation purposes (Sect. 3.2.1).

4. Withheld OCO-2 ocean glint $X_{CO_2}$ retrievals. These data could have been assimilated, but they are intentionally withheld for evaluation purposes (Sect. 3.2.1).

We first perform a simple check on the inversion results by comparing the atmospheric $CO_2$ growth rate estimated from the v10 OCO-2 MIP experiments to that derived directly from NOAA $CO_2$ measurements (Fig. 4). The growth rate is estimated from $CO_2$ measurements and model co-samples at "marine boundary layer" sites, which predominantly observe well-mixed marine boundary layer air representative of a large volume of the atmosphere. A smooth curve is then fit to these data to estimate



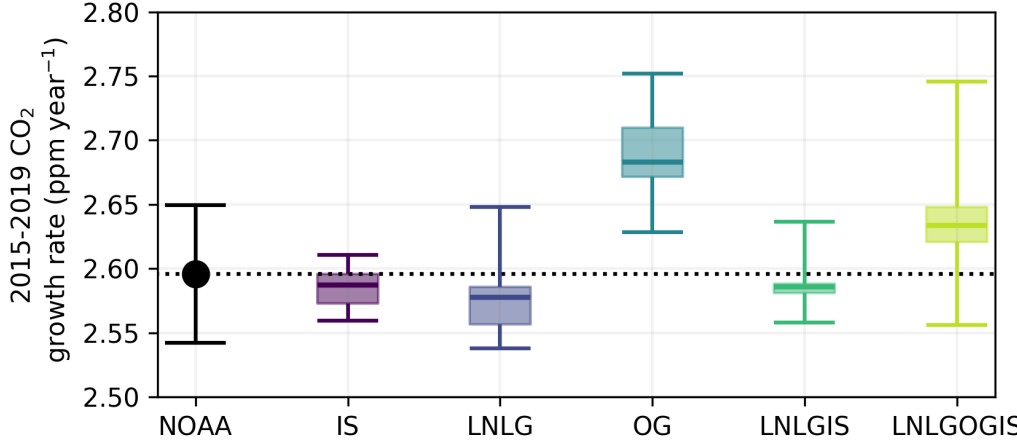

**Figure 4.** Mean 2015–2019 global mean $CO_2$ growth rate estimated from NOAA site measurements and for the v10 OCO-2 MIP experiments. The estimates of the $CO_2$ growth rate for each experiment are computed by sampling the model $CO_2$ fields at the same times and locations as those used to derive the NOAA measurement-based estimate. Each v10 OCO-2 MIP experiment is shown as a box plot, with the error bars showing the full range, the shaded region showing the interquartile range, and the solid line showing the median ensemble member of the ensemble.

the global growth rate (Thoning et al., 1989). This is the same method employed by NOAA to report the $CO_2$ growth rate (gml.noaa.gov/ccgg/trends/). We estimate the uncertainty in the measurement-based growth rate from the difference between the growth rate estimated here and that reported on the NOAA website. Differences between these estimates are primarily driven by differences in measurement sampling used for the website relative to that used here (as we are limited to withheld co-samples here). We calculate the uncertainty as the standard error of the mean for the differences between the growth rates

estimated here and by NOAA across 2015–2019. This gives an uncertainty on the 5-year growth rate of $\pm 0.053\,\mathrm{ppm\,yr^{-1}}$. Note that NOAA reports the growth rate using the X2019 scale, whereas our estimates here are from the X2007 scale, which may contribute to the differences. We find that the IS, LNLG, and LNLGIS experiments show good agreement with the NOAA estimate over this period. However, both the OG and LNLGOGIS experiments are found to have a high bias. This suggests that there may be a spurious trend in the v10 OCO-2 ocean glint $X_{CO_2}$ retrievals of 0.04–0.13 ppm yr$^{-1}$ (OG experiment bias) that

impacts flux estimates from both experiments that assimilate ocean glint data.

Second, we estimate the overall data–model agreement as the root-mean-square (RMS) error for the the withheld in situ $CO_2$, TCCON $X_{CO_2}$, withheld OCO-2 land $X_{CO_2}$, and withheld OCO-2 ocean $X_{CO_2}$ (Fig. 5). For the in situ and OCO-2 data, the normalized RMS is shown, meaning that the data–model difference is divided by the observational uncertainty (one-sigma). Overall, we find reasonably good agreement between the evaluation datasets and posterior fields for all experiments.

The OG experiment gives the largest RMS errors against the withheld in situ $CO_2$, TCCON $X_{CO_2}$, and OCO-2 land $X_{CO_2}$. This provides further evidence that the ocean glint data may have some residual biases that adversely impact the flux estimates.



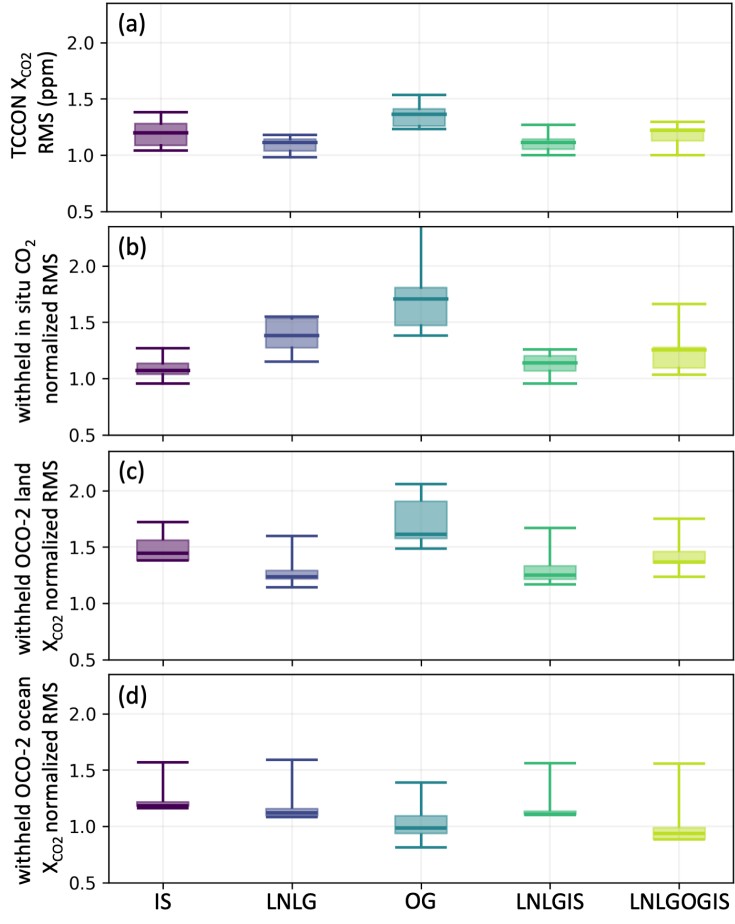

**Figure 5.** 2015–2020 root-mean-square (RMS) error between the v10 OCO-2 MIP experiments and (a) TCCON $X_{CO_2}$ retrievals, (b) withheld in situ $CO_2$ measurements, (c) withheld OCO-2 land $X_{CO_2}$ retrievals, and (d) withheld OCO-2 ocean $X_{CO_2}$ retrievals. For the comparisons with withheld in situ and OCO-2 observations, the normalized RMS estimate is plotted (that is, the data–model mismatch is divided by the observational uncertainty). Note that and NIES IS and CSU co-samples are not available and not included in this plot.

Finally, we examine the mean bias over 2015–2020 for 30° latitude bins (Fig. 6). Similar to previous comparisons, we find that the OG experiment stands out as being more biased against the independent data relative to the other experiments.
In particular, the data–model difference for the OG experiment tends to be low (higher modeled $CO_2$) than the evaluation datasets. This is particularly evident in the northern extratropics. Over 30°–60° N, where independent data is densest, we find that the OG ensemble median is biased by -0.69 ppm against TCCON, -0.74 ppm against witheld in situ, and -0.48 ppm against witheld OCO-2 LNLG, suggesting a possible meridional bias in the OCO-2 ocean $X_{CO_2}$ retrievals. The IS, LNLG, and LNLGIS experiments tend to show similar data–model differences, suggesting limited ability to distinguish between the
performance of these inversions in large-scale features.

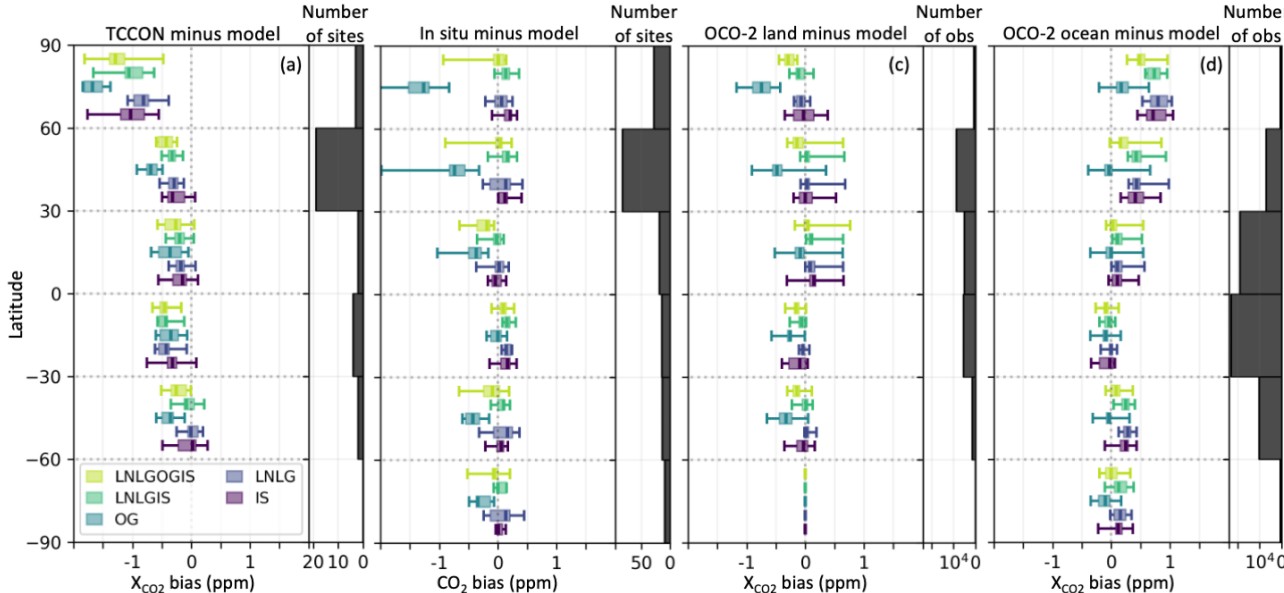

**Figure 6.** Median bias (data minus model) over 30° latitude bins averaged over 2015–2020 for (a) TCCON $X_{CO_2}$ retrievals, (b) withheld in situ $CO_2$ measurements, (c) withheld OCO-2 land $X_{CO_2}$ retrievals, and (d) withheld OCO-2 ocean $X_{CO_2}$ retrievals. Note that and NIES IS and CSU co-samples are not available and not included in this plot.

All experiments show some biases against TCCON sites. In particular, low biases (high modeled $CO_2$) are found for 0°–30° S and 60°–90° N. The underlying cause for these differences is unknown. Figure S3 shows the monthly-mean data–model differences for each TCCON site and each experiment. The differences can be quite variable between sites, but are generally similar between experiments (for IS, LNLG, and LNLGIS). Some of these differences may be related due to representativeness errors, particularly for urban sites. For example, Caltech and JPL are within Los Angeles County and show a large positive bias, while nearby Edwards is less impacted by urban emissions and shows a much smaller bias (Schuh et al., 2021). However, other differences are harder to explain, such as a negative trend in the data–model bias for Park Falls and positive at Darwin during the 2015–2020 period. Site-to-site biases among TCCON sites may also contribute to these differences.

Overall, this analysis finds that the OG experiment shows the poorest agreement against the evaluation datasets (excluding the withheld ocean glint data). The LNLGOGIS experiment shows the second worst performance against evaluation datasets, while the remaining experiments (IS, LNLG, and LNLGIS) all show good agreement against the evaluation data. These results suggest that there may be residual biases in the OCO-2 ocean glint dataset that adversely impact the OG and LNLGOGIS experiments.





## 4.2 Comparison of air–sea fluxes with pCO$_2$-based estimates

The exchange of $CO_2$ between the atmosphere and the ocean (air–sea flux) can be estimated from measurements of the surface ocean partial pressure of $CO_2$ (pCO$_2$). These pCO$_2$ data are extrapolated to global maps and combined with gas transfer velocity parameterizations to infer global maps of the air–sea $CO_2$ fluxes (Fay et al., 2021). Although significant uncertainties remain, particularly in accurately representing the gas transfer velocity (Fay et al., 2021), comparisons between the pCO$_2$-based air–sea fluxes and v10 OCO-2 MIP experiments can inform possible biases between estimates and inform potential areas

for future research.

Here, we compare v10 OCO-2 MIP air–sea fluxes to an ensemble of air–sea flux estimates from SeaFlux (Fay et al., 2021; Gregor and Fay, 2021). SeaFlux developed a standardized approach to harmonize and extend six air–sea $CO_2$ flux products from as many surface pCO$_2$ products: JENA-MLS (Rödenbeck et al., 2013), MPI-SOMFFN (Landschützer et al., 2014, 2020), CMEMS-FFN (Denvil-Sommer et al., 2019; Chau et al., 2022), CSIR-ML6 (Gregor et al., 2019), JMA-MLR (Iida et al.,

2021), and NIES-FNN (Zeng et al., 2014). For each pCO$_2$ product, we examine the mean of three air–sea fluxes obtained using different wind reanalysis datasets to estimate the gas transfer parameterization (ERA5, JRA55, and CCMP2). The spread among these six estimates provides a measure of uncertainty in the extrapolation of pCO$_2$ data to a global grid, but does not account for errors in the gas transfer velocity formulation nor the uncertainties in the reanalysis winds used as input (Fay et al., 2021). Note that the prior estimates of air–sea $CO_2$ fluxes in v10 OCO-2 MIP experiments are generally pCO$_2$-based flux

estimates, and therefore not independent from the SeaFlux datasets.

Figure 7 shows the 2015–2019 mean air–sea fluxes for each of the six SeaFlux products and for the v10 OCO-2 MIP experiments across 30° latitude bands and large ocean regions. Over the global ocean, the pCO$_2$-based air–sea fluxes tend to give stronger removals (median $= -10.0\,\mathrm{PgCO_2\,yr^{-1}}$ or $-2.7\,\mathrm{PgC\,yr^{-1}}$, range $= -0.2\,\mathrm{to}\,-12.9\,\mathrm{PgCO_2\,yr^{-1}}$ or $-3.5\,\mathrm{to}\,-2.5\,\mathrm{PgC\,yr^{-1}}$) than the v10 OCO-2 MIP, which range from $-7.9 \pm 1.9\,\mathrm{PgCO_2\,yr^{-1}}$ $(-2.1 \pm 0.5\,\mathrm{PgC\,yr^{-1}})$ for the IS experiment to

$-10.2 \pm 1.28\,\mathrm{PgCO_2\,yr^{-1}}$ $(-2.8 \pm 0.4\,\mathrm{PgC\,yr^{-1}})$ for the OG experiment. On regional scales, the v10 OCO-2 MIP experiments overlap with the pCO$_2$-based estimates except for the northern high latitudes (60°–90° N), where pCO$_2$-based estimates suggest a systematically larger removals. Similarly, the pCO$_2$-based estimates tend to give greater removals over the southern midlatitudes (20°–50° S).

The different v10 OCO-2 MIP experiments tend to give similar air–sea fluxes, except for the OG experiment in the tropics.

Although not systematic, the OG experiment suggests weaker emissions in the tropics of $0.2 \pm 1.3\,\mathrm{PgCO_2\,yr^{-1}}$ $(0.05 \pm 0.34\,\mathrm{PgC\,yr^{-1}})$ relative to the median pCO$_2$-based estimate of $1.6\,\mathrm{PgCO_2\,yr^{-1}}$ $(0.43\,\mathrm{PgC\,yr^{-1}})$ with a range of $0.4\,\mathrm{to}\,1.8\,\mathrm{PgCO_2\,yr^{-1}}$ $(0.10\,\mathrm{to}\,0.50\,\mathrm{PgC\,yr^{-1}})$. Thus, similar to the evaluation of posterior $CO_2$ fields, the OG experiment is an outlier among the v10 OCO-2 MIP experiments, further supporting the possibility that residual biases may exist in the ocean glint $X_{CO_2}$ retrievals.

## 5 Metrics for interpreting country flux estimates

To aid users in interpreting top-down country-level flux estimates, we provide two metrics. The first metric is called the "Z statistic" and quantifies the statistical agreement between the IS and LNLG NCE estimates, and thus gives an indication of





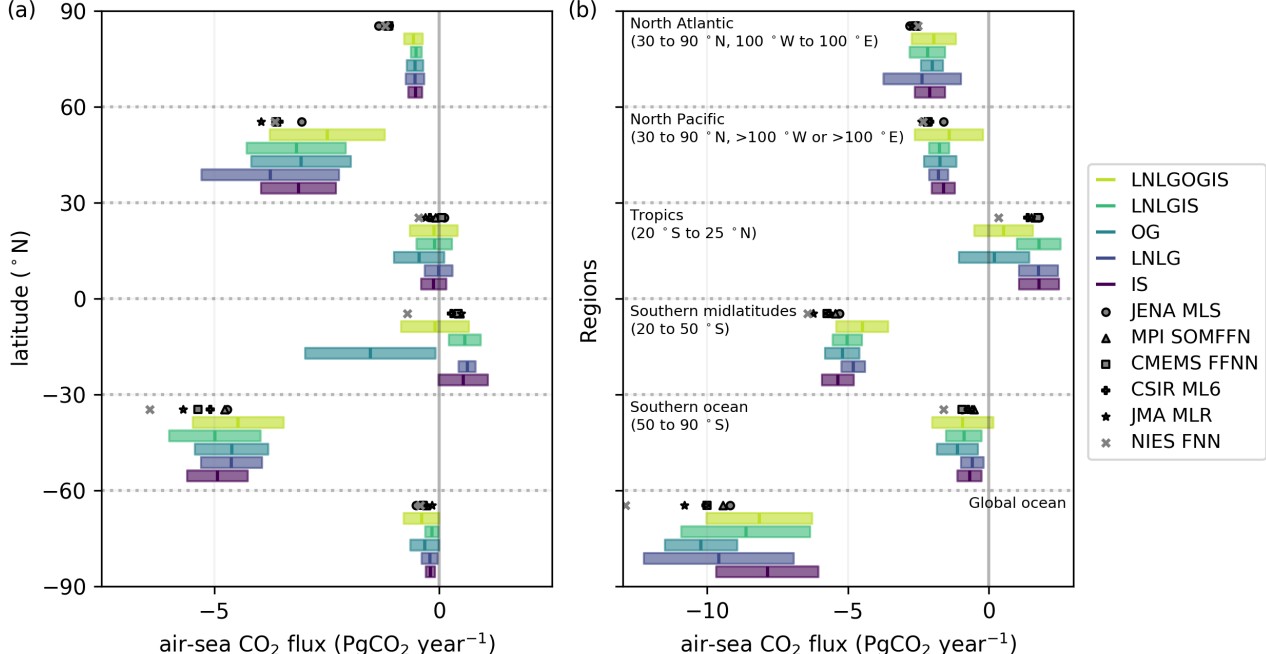

**Figure 7.** (a) Zonal-mean air–sea $CO_2$ flux (positive values represent flux towards atmosphere) for $30°$ increments of latitude based on $1° \times 1°$ estimates averaged over 2015–2019. (b) air–sea $CO_2$ flux for six large ocean regions. Colored bars show the MIP experiment results (median +/- one standard deviation) and the symbols show the $pCO_2$-based air–sea fluxes from the six SeaFlux products.

how robust flux estimates are across the v10 OCO-2 MIP experiments (Sect. 5.1). The second metric is called the Fractional Uncertainty Reduction (FUR) and informs the impact of the assimilated $CO_2$ data on the estimated fluxes (Sect. 5.2).

### 5.1 Z statistic

The Z statistic is defined as,

$$\text{Z statistic} = \frac{\overline{\text{NCE}_{\text{LNLG}}} - \overline{\text{NCE}_{\text{IS}}}}{std(\text{NCE}_{\text{LNLG}} - \text{NCE}_{\text{IS}})}, \tag{7}$$

where the denominator represents the standard deviation in $\text{NCE}_{\text{LNLG}} - \text{NCE}_{\text{IS}}$ across the ensemble members. Differences in NCE and $\Delta C_{\text{loss}}$ between v10 OCO-2 MIP experiments can be considerable. As an example, Fig. 8a shows that differences between $\overline{\text{NCE}_{\text{LNLG}}}$ and $\overline{\text{NCE}_{\text{IS}}}$ are notable for South America and Africa. The LNLG experiment gives more positive $\Delta C_{\text{loss}}$

(carbon loss from land) over northern sub-Saharan Africa and northeast South America, but more negative $\Delta C_{\text{loss}}$ over southern tropical Africa, southern and eastern South America, and southeast Asia. We examine the Z statistic (Fig. 8b) to quantify the statistical significance of these difference (magnitude greater than 1.96 indicates statistically significant differences at level $\alpha = 0.05$). Most countries do not have statistically significant differences, indicating relatively good agreement between the IS

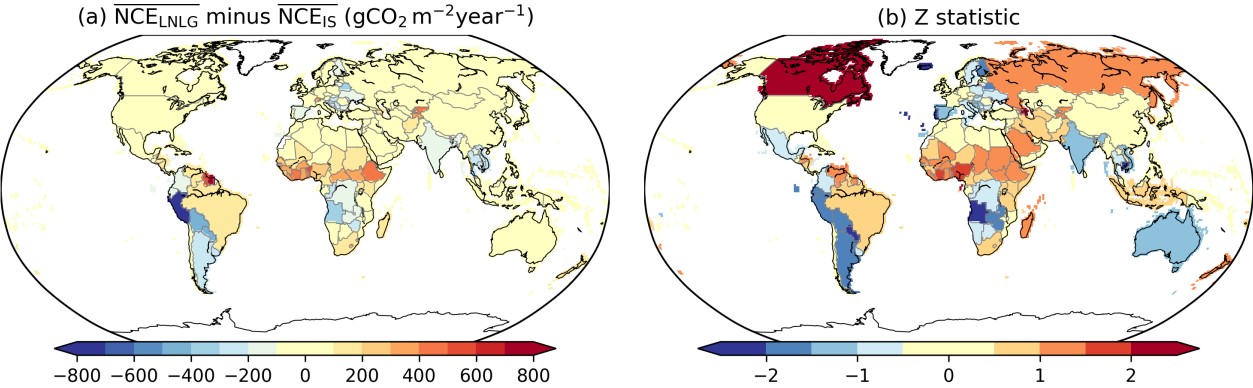

**Figure 8.** Difference between LNLG and IS experiments. (a) $\overline{\mathrm{NCE_{LNLG}}}$ minus $\overline{\mathrm{NCE_{IS}}}$, and (b) The Z statistic (Eqn. 7) indicating the difference between LNLG and IS experiments.

and LNLG ensembles. Significant differences primarily occur in small to mid-sized tropical countries. Canada also shows a

systematic difference driven by small uncertainties in the IS and LNLG estimates.

## 5.2 Fractional uncertainty reduction (FUR)

Byrne et al. (2022) reports the uncertainty in NCE as the standard deviation across v10 OCO-2 MIP ensemble members (estimated using Eqn. 4). This metric incorporates uncertainties related to model transport and aspects of the inversion configuration, such as optimization technique and a priori flux estimates. However, this metric is different to the uncertainty metric

usually computed in a Bayesian framework, that is, the Bayesian posterior uncertainty. That uncertainty quantifies the impact of errors in the observations and prior constraints on the posterior flux estimates. The Bayesian posterior uncertainty is not reported for practical reasons, as the majority of contributing models do not calculate this quantity, so it is not possible to calculate this quantity across the ensemble.

    In this section, we examine the posterior uncertainty estimates from two contributing inversion systems (CAMS and TM5-

4DVar) and compare these estimates to the ensemble-based uncertainty estimate provided with the dataset. Then, we define the metric of Fractional Uncertainty Reduction (FUR) between the posterior and prior NCE estimates based on the TM5-4DVar model (as CAMS does not estimate uncertainties for the LNLGIS and LNLGOGIS experiments), which can be used to understand the relative impact of assimilated atmospheric $CO_2$ data on estimates of country-level NCE and $\Delta C_{\mathrm{loss}}$.

    Both CAMS and TM5-4DVar estimate $CO_2$ fluxes using four-dimensional variational assimilation (4D-Var) and estimate

posterior uncertainty estimates using a Monte Carlo method derived by Chevallier et al. (2007). The realism of the prior and posterior CAMS uncertainty estimates have already been the topic of several studies (see Chevallier, 2021, and references therein). Figure 9 shows the ensemble-based uncertainty, prior/posterior uncertainty from CAMS (prior, IS and LNLG only) and prior/posterior uncertainty from TM5-4DVar for four countries. Notably, the magnitudes of the prior/posterior uncertainties from CAMS and TM5-4DVar are quite different, with CAMS uncertainties being 2–8 times larger. Differences in prior/poste-

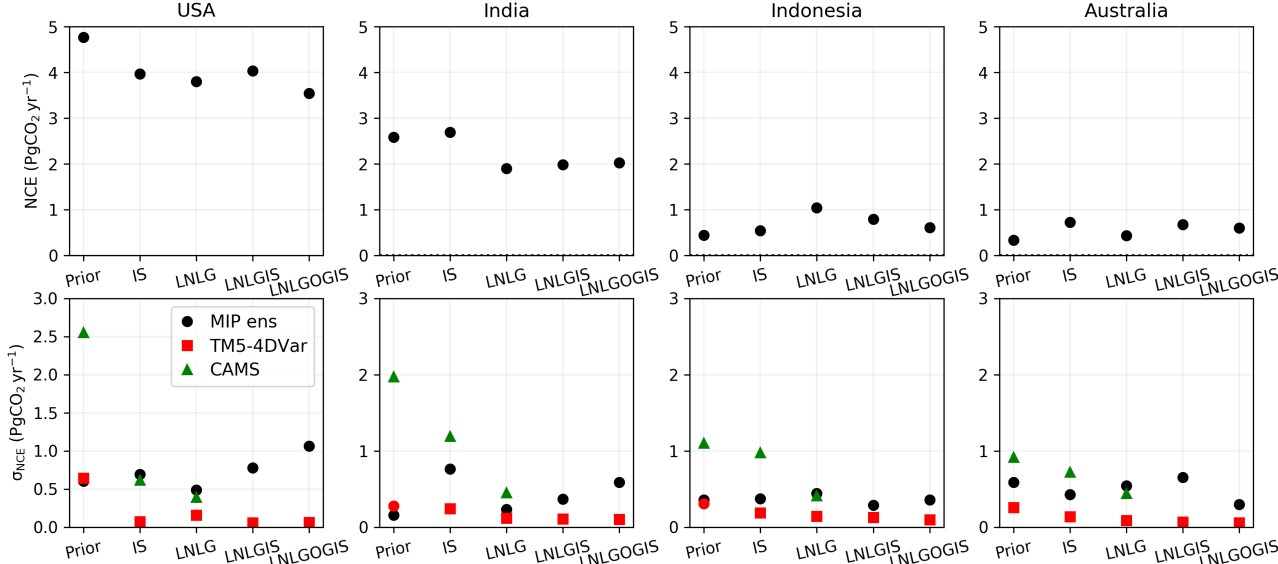

**Figure 9.** (top) $\overline{NCE}$ and (bottom) $\sigma_{NCE}$ for four countries in 2018. The v10 OCO-2 MIP ensemble spread-based error estimate is shown in black, the TM5-4DVar Bayesian uncertainty estimate is shown in red, and the CAMS Bayesian uncertainty estimate is shown in green (only for Prior, IS, and LNLG).

rior uncertainties of this magnitude are not unusual among inversion systems, and highlight the sensitivity of Bayesian uncertainty estimates to choices about prior uncertainties. Both CAMS and TM5-4DVar posterior uncertainties are smaller relative to their prior by similar amounts, driven by the assimilated $CO_2$ data. The magnitude of the ensemble-based uncertainty tends to fall in-between the CAMS and TM5-4DVar estimates. However, the CAMS and TM5-4DVar posterior uncertainty estimates decrease as more data are assimilated (as expected), while the ensemble spread does not. In fact, the ensemble spread increases

with data density in some cases (e.g., Australia LNLGIS). Thus, overall, we find that the ensemble-based uncertainty estimate is of similar magnitude to the prior/posterior estimate, but that the magnitude of posterior uncertainty is quite dependent on the assumed prior uncertainty.

    We now calculate the FUR metric in NCE from the TM5-4DVar Bayesian uncertainties (note that we use TM5-4DVar only because CAMS does not report LNLGIS or LNLGOGIS uncertainties). FUR is calculated from the prior flux standard deviation

($\sigma_{\text{prior}}$) and posterior flux standard deviation ($\sigma_{\text{posterior}}$) as:

$$\text{FUR} = 1 - \frac{\sigma_{\text{posterior}}}{\sigma_{\text{prior}}} \qquad (8)$$

This quantity ranges between 0 and 1, with larger values indicating that the Bayesian uncertainties have decreased more (relative to the prior) due to the observational constraints from assimilated data. This metric is useful for understanding how the assimilation of data influences the NCE and $\Delta C_{\text{loss}}$ estimates, which may not be captured by the ensemble spread. For

example, Saudi Arabia has a small NCE uncertainty estimate but this is largely driven by prior knowledge that biosphere $CO_2$ fluxes and the atmospheric $CO_2$ data has little impact on the NCE estimate.

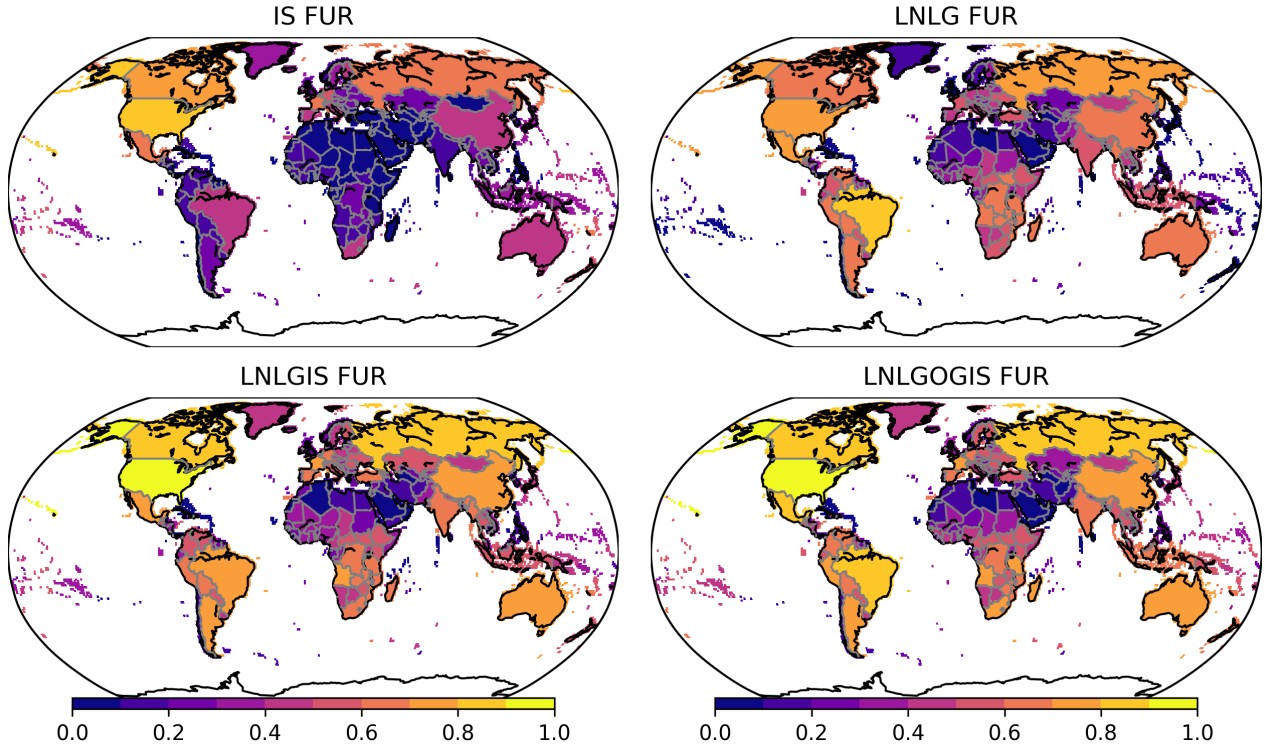

**Figure 10.** Estimate of the Fractional Uncertainty Reduction (FUR) on the v10 OCO-2 MIP estimates for each experiment based on Bayesian uncertainty estimates from the TM5-4DVar inversion.

Figure 10 shows FUR for the IS, LNLG, LNLGIS, and LNLGOGIS experiments. FUR is larger in regions with denser observational coverage. For example, the IS FUR is close to 1 in the USA and parts of Europe, reflecting dense $CO_2$ measurements, but it remains small for many tropical countries, where sampling is sparse. Meanwhile, the LNLG experiment generally has larger FUR values than the IS experiment in the tropics, reflecting denser sampling, but has lower values for some small high-latitude countries, such as in Scandinavia.

## 6 Dataset description

The dataset described in this paper, Byrne et al. (2022), provides annual totals of country-level and $1° \times 1°$ gridded $\Delta C_{\text{loss}}$, NBE, NCE, $F_{\text{rivers export}}$, and the combined $F_{\text{crop trade}} + F_{\text{wood trade}}$ fluxes, as well as their uncertainties over 2015–2020. In addition, the country-level Z statistic (Eqn. 7) and FUR (Eqn. 8) metrics are provided to help interpret the flux and stockchange estimates. These data are provided for the v10 OCO-2 MIP IS, LNLG, LNLGIS, and LNLGOGIS experiments. The OG experiment is excluded due to poor evaluation against independent $CO_2$ measurements and pCO$_2$-based air–sea fluxes, likely due to residual $X_{CO_2}$ biases in the OCO-2 ocean glint $X_{CO_2}$ retrievals (Sect. 4). We note that biases in ocean glint $X_{CO_2}$ retrievals will also adversely impact flux estimates from the LNLGOGIS, and caution against using these data when they show



differences from the IS, LNLG, and LNLGIS experiments. Future improvements to the OCO-2 $X_{CO_2}$ retrievals are expected
to reduce residual $X_{CO_2}$ biases and thus the quality of the LNLGOGIS experiment is expected to improve in future OCO-2
MIP experiments.

For the $1° \times 1°$ gridded dataset, we emphasize that caution is needed in interpreting these data. As discussed in Sect. 1.3,
atmospheric $CO_2$ inversion analyses provide the best constraints on the largest spatial scales (e.g., continental-to-global). The
confidence in these top-down estimates decreases at smaller spatial scales. The minimum spatial resolution for robust flux
estimates is dependent on the density and precision of the measurements, and is challenging to quantify. However, scales
smaller than France or Germany in geographic extent are unlikely to be meaningfully constrained. Thus, we recommend
only using $1° \times 1°$ $CO_2$ fluxes aggregated to larger spatial scales. In aggregating, we recommend propagating uncertainties
by assuming first 100% correlation (sum of the $1° \times 1°$ uncertainties) and then 0% correlation (square root of the sum of
the squared uncertainties) between grid cells. We strongly encourage contacting the authors before using the gridded $1° \times 1°$
dataset.

These data are available for download from the Committee on Earth Observation Satellites' (CEOS) website:
https://doi.org/10.48588/npf6-sw92. The country-level data are available for download as comma-separated values (CSV),
Network Common Data Form (NetCDF) and Microsoft Excel worksheet files. The $1° \times 1°$ gridded dataset is available as a
NetCDF file.

## 7 Characteristics of the dataset

Globally, over 2015–2020, we report FF emissions of $35.79 \pm 1.50\,\mathrm{PgCO_2\,yr^{-1}}$ ($9.76 \pm 0.41\,\mathrm{PgC\,yr^{-1}}$), $F_{\mathrm{rivers\,export}}$ of
$-3.35 \pm 0.59\,\mathrm{PgCO_2\,yr^{-1}}$ ($-0.91 \pm 0.16\,\mathrm{PgC\,yr^{-1}}$), and globally balanced $F_{\mathrm{crop\,trade}}$ and $F_{\mathrm{wood\,trade}}$. Table 3 gives the
global annual mean changes in the atmospheric burden of $CO_2$, $\Delta C_{\mathrm{gain}}$ and ocean sequestration. Across the experiments, the
median fraction of fossil fuel emissions remaining in the atmosphere is 55–56%, while 32–36% is sequestered by the ocean and
9-13% is sequestered by terrestrial ecosystems. Note that this omits land use change (LUC) emissions of $\sim 3.85\,\mathrm{PgCO_2\,yr^{-1}}$
($\sim 1.05\,\mathrm{PgC\,yr^{-1}}$, Friedlingstein et al., 2022), which are compensated for by additional carbon uptake by land. Of the com-
bined FF+LUC emissions, 50% remains in the atmosphere, 29–33% is sequestered by the ocean and 18-21% is sequestered
by terrestrial ecosystems. Relative to Global Carbon Budget 2021 (GCB 2021; Friedlingstein et al., 2022) we find 2.24–
$3.53\,\mathrm{PgCO_2\,yr^{-1}}$ ($0.61$–$0.96\,\mathrm{PgC\,yr^{-1}}$) less removal by land (mean/median difference) but greater removal by the ocean of
$0.87$–$2.24\,\mathrm{PgCO_2\,yr^{-1}}$ ($0.24$–$0.61\,\mathrm{PgC\,yr^{-1}}$), however, these difference are consistent within one standard deviation of the
mean/median values. Interestingly, we report greater removals by the ocean than GCB 2021 but reduced air–sea flux relative
to SeaFlux. This can be explained by the fact that $pCO_2$-based air–sea flux estimates generally give larger mean ocean carbon
uptake than model estimates (Fay and McKinley, 2021) and that we estimate a larger $F_{\mathrm{rivers\,export}}$ than GCB 2021.

Meridionally, NCE is largest in the northern extratropics, coinciding with the largest FF emissions (Fig. 11). However, the
northern extratropics also show negative $\Delta C_{\mathrm{loss}}$, implying increasing terrestrial carbon stocks, particularly between 30°–60° N.
NCE is less positive in the tropics, primarily due to lower FF emissions. However, this region tends to show neutral-to-positive



**Table 3.** 2015-2020 global mean atmospheric increase, terrestrial carbon gain ($\Delta C_{\text{gain}}$) and ocean carbon gain from the IS, LNLG, LNLGIS, and LNLGOGIS experiments (mean/median $\pm$ one standard deviation). Positive values of $\Delta C_{\text{gain}}$ and ocean carbon gain indicate increases in carbon stocks. GCB 2021 were obtained from the Global Carbon Budget 2021 (Friedlingstein et al., 2022) with $\Delta C_{\text{gain}}$ calculated as the difference between the land sink and land-use change emissions with errors propagated in quadrature.

| Experiment | Atmosphere | $\Delta C_{\text{gain}}$ | Ocean carbon gain |
|---|---|---|---|
| IS | $19.73 \pm 0.19 \, \text{PgCO}_2 \text{yr}^{-1}$ <br> $(5.38 \pm 0.05 \, \text{PgC yr}^{-1})$ | $4.58 \pm 2.44 \, \text{PgCO}_2 \text{yr}^{-1}$ <br> $(1.25 \pm 0.66 \, \text{PgC yr}^{-1})$ | $11.35 \pm 2.01 \, \text{PgCO}_2 \text{yr}^{-1}$ <br> $(3.10 \pm 0.55 \, \text{PgC yr}^{-1})$ |
| LNLG | $19.64 \pm 0.09 \, \text{PgCO}_2 \text{yr}^{-1}$ <br> $(5.36 \pm 0.02 \, \text{PgC yr}^{-1})$ | $3.29 \pm 2.93 \, \text{PgCO}_2 \text{yr}^{-1}$ <br> $(0.90 \pm 0.80 \, \text{PgC yr}^{-1})$ | $12.91 \pm 2.63 \, \text{PgCO}_2 \text{yr}^{-1}$ <br> $(3.52 \pm 0.72 \, \text{PgC yr}^{-1})$ |
| LNLGIS | $19.64 \pm 0.06 \, \text{PgCO}_2 \text{yr}^{-1}$ <br> $(5.36 \pm 0.02 \, \text{PgC yr}^{-1})$ | $4.19 \pm 2.77 \, \text{PgCO}_2 \text{yr}^{-1}$ <br> $(1.14 \pm 0.75 \, \text{PgC yr}^{-1})$ | $11.98 \pm 2.32 \, \text{PgCO}_2 \text{yr}^{-1}$ <br> $(3.27 \pm 0.64 \, \text{PgC yr}^{-1})$ |
| LNLGOGIS | $19.97 \pm 0.18 \, \text{PgCO}_2 \text{yr}^{-1}$ <br> $(5.45 \pm 0.05 \, \text{PgC yr}^{-1})$ | $4.03 \pm 2.36 \, \text{PgCO}_2 \text{yr}^{-1}$ <br> $(1.10 \pm 0.64 \, \text{PgC yr}^{-1})$ | $11.54 \pm 1.79 \, \text{PgCO}_2 \text{yr}^{-1}$ <br> $(3.15 \pm 0.49 \, \text{PgC yr}^{-1})$ |
| GCB 2021 | $19.8 \pm 0.73 \, \text{PgCO}_2 \text{yr}^{-1}$ <br> $(5.39 \pm 0.2 \, \text{PgC yr}^{-1})$ | $6.82 \pm 3.15 \, \text{PgCO}_2 \text{yr}^{-1}$ <br> $(1.86 \pm 0.86 \, \text{PgC yr}^{-1})$ | $10.67 \pm 1.83 \, \text{PgCO}_2 \text{yr}^{-1}$ <br> $(2.91 \pm 0.5 \, \text{PgC yr}^{-1})$ |

$\Delta C_{\text{loss}}$, suggesting that terrestrial carbon stocks may be decreasing. The LNLG and IS results also differ most in the tropics, with LNLG suggesting greater terrestrial carbon stock loss over 0°–30° N but less over 0°–30° S. The differences in CO$_2$ fluxes
between these experiments are not well understood, and both experiments evaluate well against independent data (Sect. 4).

The spatial distribution of NCE over 2015–2020 at $1° \times 1°$ and aggregated to country-scale for the LNLGIS experiment is shown in Fig. 12. At $1° \times 1°$ (Fig. 12a-b), localized fossil fuel emissions are visible, generally corresponding to urban areas and industrialized regions. These emissions are interspersed over broad source and sink structures that are driven by biosphere removals or emissions. Land biosphere removal is most evident across the northern mid-high latitudes. In contrast, tropical
removals and emissions are more regional. When NCE is aggregated to the country-scale (Fig. 12c-d), most countries are net sources driven by fossil fuel emissions, particularly in the northern extratropics. Figure 12e-f shows the 2015–2020 mean country-level $\Delta C_{\text{loss}}$ for the LNLGIS experiment. Increasing terrestrial carbon stocks (negative $\Delta C_{\text{loss}}$) is found for most extratropical countries, while tropical countries can have gains or losses. Notably, the uncertainty in $\Delta C_{\text{loss}}$ is larger in the tropics, particularly for mid-sized countries. Overall, small to mid-sized countries generally have uncertainties comparable to
the magnitude of $\Delta C_{\text{loss}}$, reflecting the fact that atmospheric CO$_2$ measurements best constrain fluxes over large scales. Spatial maps of NCE and $\Delta C_{\text{loss}}$ for each experiment are shown in the supplementary materials (Fig. S4-7).

Differences in NCE and $\Delta C_{\text{loss}}$ between the v10 OCO-2 MIP experiments can be considerable (the statistical significance of these differences is quantified by the Z statistic, see Sect. 5.1). The underlying cause of differences between the v10 OCO-2 MIP experiments are not well understood, but the differences are likely impacted by the different spatial and temporal distribution of

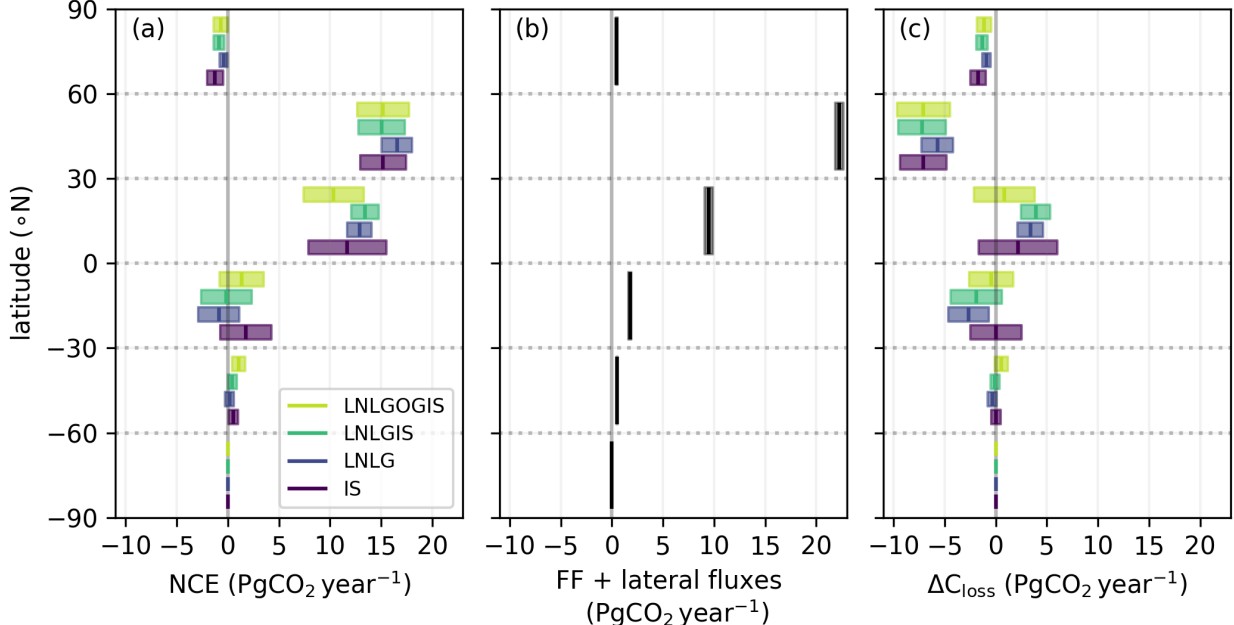

**Figure 11.** Zonal-mean (a) NCE, (b) FF + lateral fluxes, and (c) $\Delta C_{\text{loss}}$ for $30°$ increments of latitude based on $1° \times 1°$ estimates averaged over 2015–2020. IS, LNLG, LNLGIS and LNLGOGIS median estimates are shown by solid lines and one-sigma uncertainties are shown by the shaded region.

LNLG and IS measurements (see Sec. 5.2), model transport errors (Stephens et al., 2007; Schuh et al., 2019, 2022) and residual retrieval biases in the OCO-2 $X_{\text{CO}_2}$ retrievals (Peiro et al., 2022). Unfortunately, the regions showing the largest differences in fluxes generally have few independent atmospheric $CO_2$ measurements for validation, limiting our ability to distinguish between different causes. Thus, we believe that NCE and $\Delta C_{\text{loss}}$ estimates are most reliable when agreement is found across the v10 OCO-2 MIP experiments.

We will now show examples of carbon budgets for four countries from this dataset. Figure 13 shows the 2015–2020 mean FF, $F_{\text{rivers export}}$, $F_{\text{crop trade}}$, $F_{\text{wood trade}}$, $\Delta C_{\text{loss}}$, and NCE fluxes for the USA, India, Indonesia, and Australia. All of the $CO_2$ fluxes on the left of the dashed line combine to give the NCE flux constrained by the v10 OCO-2 MIP experiments. We find that FF is the strongest contributor to NCE for all countries, but that $\Delta C_{\text{loss}}$ also plays a strong modulating role. For example, negative $\Delta C_{\text{loss}}$ (increasing terrestrial carbon stocks) for the USA reduces NCE to be less than would be expected
given the FF emissions. Conversely, Indonesia has positive $\Delta C_{\text{loss}}$ (decreasing terrestrial carbon stocks), resulting in increased NCE relative to FF. Some countries also show differences in $\Delta C_{\text{loss}}$ between v10 OCO-2 MIP experiments. For example, the LNLG and LNLGIS experiments suggest negative $\Delta C_{\text{loss}}$ for India, while the IS suggest $\Delta C_{\text{loss}}$ is roughly neutral. Figures of carbon budgets for 28 additional countries (Fig. S8) and 14 regions (Fig. S9) are shown in the supplementary materials.

The carbon budgets can also be examined for individual years (Fig. 14). Both Indonesia and Australia show considerable
variations in $\Delta C_{\text{loss}}$ that drive variations in NCE over this period. Indonesia has a large positive $\Delta C_{\text{loss}}$ in 2015, driven

**Figure 12.** Median ($\overline{\mathrm{NCE}}$) and one standard deviation ($\sigma_{\mathrm{NCE}}$) of NCE on a (a-b) $1° \times 1°$ grid and (c-d) aggregated to country-scale for the v10 OCO-2 MIP LNLGIS experiment averaged over 2015–2020. (e-f) Median and one standard deviation of country-scale $\Delta C_{\mathrm{loss}}$ averaged over 2015–2020 derived from the LNLGIS v10 OCO-2 MIP experiment.

by warm-dry weather and fires during 2015 El Niño (Yin et al., 2016). Australia showed strong negative $\Delta C_{\mathrm{loss}}$ (except for IS) during 2016, which was the 15th wettest year on record (precipitation 17% above average; Bureau Of Meteorology,

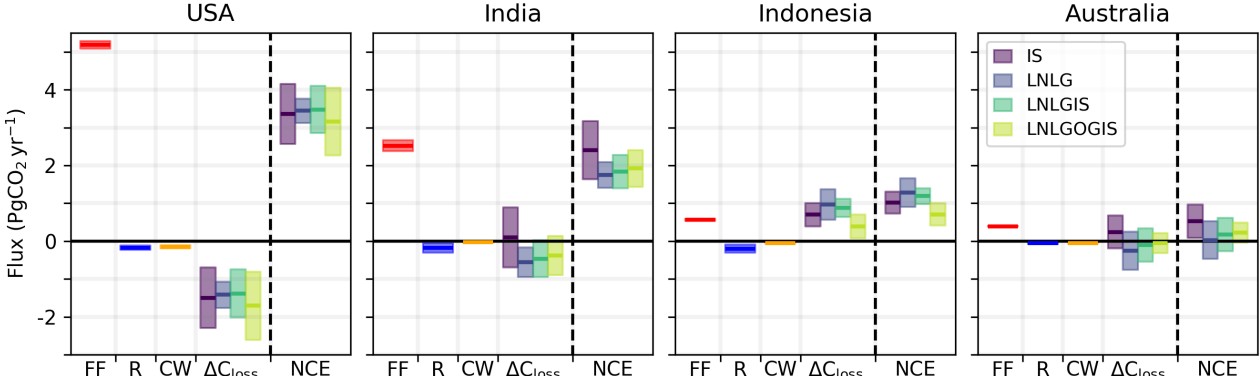

**Figure 13.** $CO_2$ budget for the USA, India, Indonesia, and Australia averaged over 2015–2020. Bars show the median +/- one standard deviation of FF, $F_{\text{rivers export}}$ (R), $F_{\text{crop trade}}+F_{\text{wood trade}}$ (CW), $\Delta C_{\text{loss}}$, and NCE (note that these quantities are related through Eqn. 2).

2017). Australia also showed anomalous positive $\Delta C_{\text{loss}}$ during 2019, which was the warmest and driest year on record, with considerable terrestrial carbon loss related to biomass burning in the southeast (Byrne et al., 2021). Variations in NCE are also

found related to FF emissions. In particular, a reduction in NCE is found for 2019 and 2020 in the USA that is primarily linked to a reduction in FF emissions rather than $\Delta C_{\text{loss}}$. Timeseries of NCE and $\Delta C_{\text{loss}}$ for 28 additional countries (Fig. S10, S11) and 14 regions (Fig. S12, S13) are shown in the supplementary materials.

## 8 Discussion

Here we discuss the current limitations of top-down country-level $CO_2$ budgets and activities that can improve these estimates.

Sect. 8.1 discusses current $CO_2$ observing systems and possible future expansions. Sect. 8.2 discusses current atmospheric $CO_2$ inversion systems, planned developments, and opportunities for improvement. Finally, Sect. 8.3 discusses remaining challenges in estimating carbon stock changes from atmospheric $CO_2$ inversions.

### 8.1 Observations

In the context of global inversion analyses, measurements of atmospheric $CO_2$ best inform annual-mean biosphere–atmosphere

$CO_2$ fluxes over large spatial scales (e.g., continental-to-global) due to rapid mixing in the atmosphere and gaps in current measurement coverage. The confidence in these top-down estimates decreases as we move to smaller spatial scales, with the minimum spatial scale being dependent on the density, precision and sensitivity of the measurements. Future refinements in top-down $CO_2$ budgets will depend on increasing observational density (Sect. 8.1.1) , improved validation (Sect. 8.1.2), and data harmonization (Sect. 8.1.3).

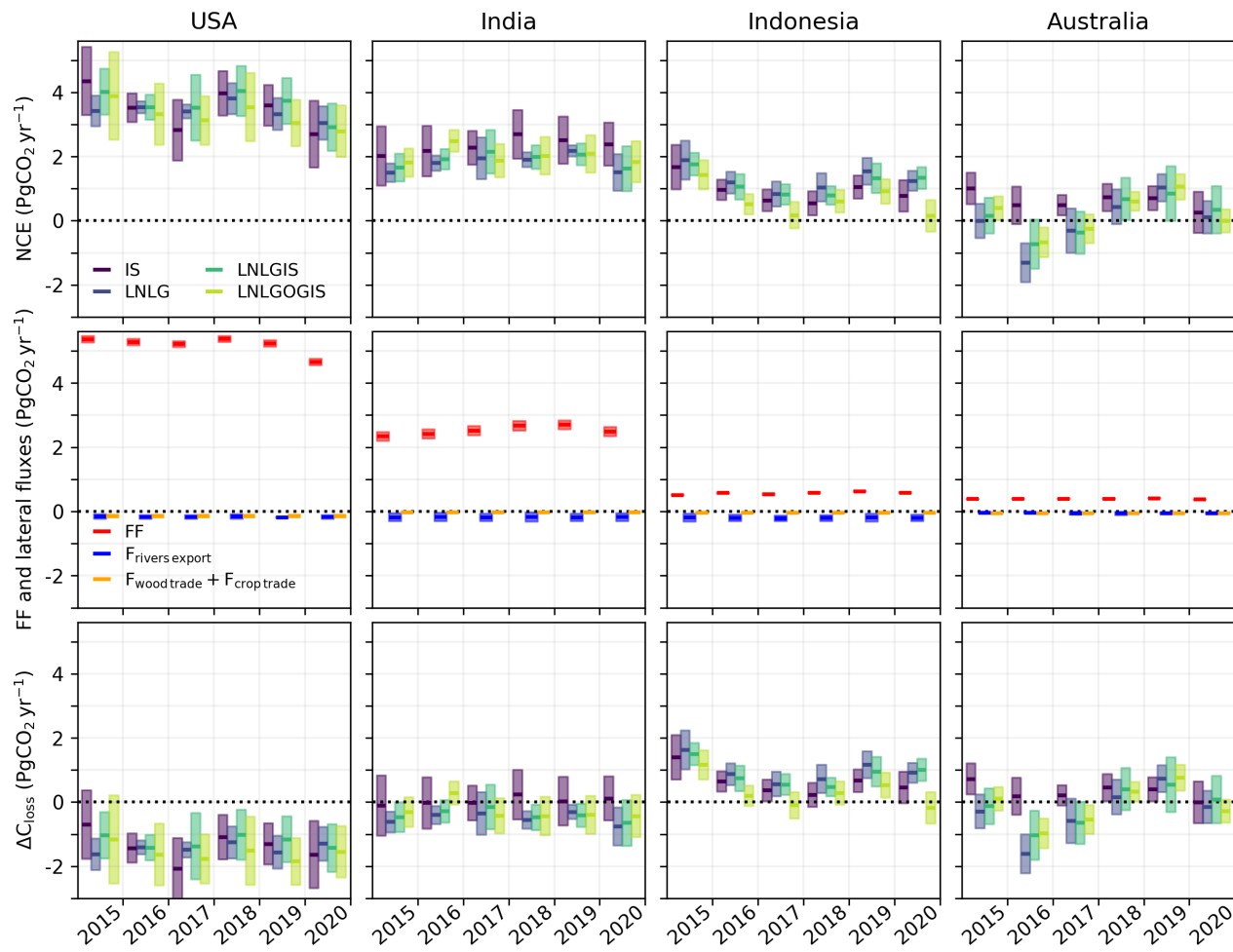

**Figure 14.** Timeseries of the carbon budget for the USA, India, Indonesia, and Australia. Solid lines show the median estimates and shaded areas show +/- one standard deviation.

### 8.1.1 Expanding observations

An expanding network of $CO_2$ observing systems provides an opportunity to reduce uncertainties in top-down estimates of NCE. Across much of the globe, country-level estimates of NCE have been limited by the observational coverage of in situ $CO_2$ measurements and $X_{CO_2}$ retrievals. However, there are a number of planned expansions in observing systems that will help fill data gaps.

The first-generation of space-based $CO_2$ systems currently in operation (GOSAT, GOSAT-2, OCO-2, OCO-3, TanSat) were designed primarily as proof-of-concept missions to demonstrate that space-based measurements could yield $X_{CO_2}$ retrievals with the precision and accuracy required to quantify emissions and removals of $CO_2$. Planned future missions will expand



and improve upon current observing systems. MicroCarb, a France-UK mission, is expected to start operations in 2023 with an additional spectral band to better characterize the light path for the estimation of $X_{CO_2}$ (Bertaux et al., 2020). Japan's

GOSAT-GW mission (https://gosat-gw.nies.go.jp/en/), which will be launched in early 2024, will also incorporate improved capabilities for $CO_2$ as well as $CH_4$. Soon after, NASA plans to launch the GeoCarb mission (https://www.ou.edu/geocarb), which will be hosted on a commercial communications satellite in geostationary orbit at a longitude around $85°$ W. From that vantage point, GeoCarb can return the data needed to estimate the column average dry air mole fraction of $CO_2$, $CH_4$ and carbon monoxide (CO) over most of North and South America at a spatial resolution of 5 to 10 km every day. In 2025, the

European Copernicus program will begin to deploy the first operational $CO_2$ and $CH_4$ monitoring constellation, CO2M (Pinty et al., 2017; Janssens-Maenhout et al., 2020). The CO2M constellation will eventually include up to three satellites, flying in formation to collect measurements at 2 km by 2 km resolution over the entire globe at weekly intervals. In addition, a follow-on to the Chinese TanSat mission is currently under development (Yang et al., 2018).

Most current and planned space-based $CO_2$ observing systems are passive, in that they rely on reflected sunlight to retrieve

$X_{CO_2}$. Active satellite missions, which use lidars for their light source, could provide coverage when reflected sunlight is not available or of insufficient intensity, such as at night and at high latitudes in the winter hemisphere when solar zenith angles are large. These systems also have the potential to better characterize systematic errors in current passive instruments by using pulse timing information to get a better estimate of path length and to filter out scattered light from clouds and aerosols (Abshire et al., 2010).

As space-based $CO_2$ observing systems expand, sub-orbital discrete air sampling (i.e., flask) and continuous $CO_2$ observing systems will remain critical for developing top-down $CO_2$ budgets. These in situ observations are the global standard for GHG measurements, because they can undergo direct calibration relative to the WMO $CO_2$-in-air mole fraction scale, which is SI-traceable (Hall et al., 2021). , In contrast, open-path remote sensing measurements (both TCCON and satellite) can not be calibrated using standard gasses; they can only be compared to in situ vertical profile observations made relative to the WMO

scale, with the differences used to adjust the remote sensing observations (e.g., Wunch et al., 2011). As such, in situ data are critical for linking remote sensing observations of $CO_2$ to the accepted trace gas scales. In situ data also provide complementary observational coverage to space-based observing systems (Byrne et al., 2017). Space-based measurements have broad spatial coverage but with seasonal variations driven by sunlight, and have data gaps in persistently cloudy regions. In contrast, flask and in situ data can be deployed year-round and regardless of cloud cover. Additionally, in situ observations most typically

represent the planetary boundary layer where flux signals in atmospheric $CO_2$ are larger than the signal as expressed in the column mean (Feng et al., 2019). Thus, these data play a critical role for improving carbon cycle constraints, especially in high latitude and persistently cloudy regions (such as the tropics), and we encourage an expansion of these systems in these undersampled regions. Regular measurements of $CO_2$ using light aircraft above several sites in Amazonia exist (e.g., Gatti et al., 2021; Miller et al., 2021), but these measurement records, as well as a nascent aircraft program in Uganda, have been so

far funded using short-term grants.

Measurements of stable- ($^{13}C/^{12}C$) and radio- ($^{14}C/C$) isotope ratios of carbon in $CO_2$ provide powerful tools for source attribution. Radiocarbon is absent from fossil fuels making it ideal for distinguishing fossil versus biologic carbon fluxes, and



inversions using measurements of $CO_2$ and $^{14}C/C$ have been used to provide top-down constraints on national-scale fossil $CO_2$ emissions (Basu et al., 2020). Atmospheric $^{13}C/^{12}C$ ratios provide insight into ecosystem stress and its relation to climate via

constraint of ecosystem water use efficiency (photosynthesis relative to water loss by transpiration) and has been used in box models (Keeling et al., 2017) and inversions (Peters et al., 2018). Atmospheric $^{13}C/^{12}C$ ratio data are generally available where discrete air samples are collected by various networks, but $^{14}C/C$ ratio data are more limited as they tend to require larger samples and measurement costs are greater. Other tracers closely related to $CO_2$, such as $O_2/N_2$ (Keeling and Graven, 2021) and Carbonyl Sulphide (e.g., Hu et al., 2021; Remaud et al., 2022) are also limited yet provide valuable information on global

ocean/NBE and regional-scale photosynthesis/respiration partitioning, respectively. Increasing the temporal and spatial density of these data, particularly across poorly sampled regions, will allow for more diagnostic power of carbon cycle processes than is possible with $CO_2$ alone.

### 8.1.2   Data validation

Validation of $X_{CO_2}$ retrievals is critical for ensuring that retrieval biases do not strongly impact flux estimates. Current gaps in

coverage of ground-based and airborne measurements have limited our confidence in flux inferences from space-based data. For example, large $CO_2$ emissions over northern Sub-Saharan Africa are a robust feature of the inversions that assimilate satellite $X_{CO_2}$ retrievals (Palmer et al., 2019), but there are few independent $CO_2$ measurements to confirm whether this inference is a real signal or an artifact of regional retrieval biases. Increased validation of space-based observations will also provide critical support for improved space-based inferences. Space-based measurements rely on validation against ground

based $X_{CO_2}$ retrievals from the TCCON (Wunch et al., 2011) and the COllaborative Carbon Column Observing Network (COCCON, Frey et al., 2019). In turn, these sites rely on in situ $CO_2$ measurements from aircraft profiles and AirCore (Karion et al., 2010) to tie their measurements to the WMO scale (Wunch et al., 2010; Messerschmidt et al., 2011). These data have been critical for validating and improving $X_{CO_2}$ retrievals (Wunch et al., 2017b; O'Dell et al., 2018; Kiel et al., 2019). Continued funding of these activities will be crucial for improving top-down $CO_2$ flux estimates and expansion of these observing systems

into undersampled regions, such as the tropics and high latitudes, will also be important for identifying and addressing residual $X_{CO_2}$ retrieval biases. In addition, efforts to cross-calibrate TCCON and COCCON sites will be helpful for minimizing site-to-site biases and identifying spurious drifts in $X_{CO_2}$. We encourage future campaigns aimed at site-to-site comparisons similar to the FRM4GHG campaign that deployed total column GHG traveling standard instruments at several TCCON sites as part of ESA's FRM4GHG-2 project (Sha et al., 2020).

### 685   8.1.3   Data harmonization

Further advancements in top-down flux estimates will be possible through combining the observational constraints from the constellation space-based sensors and ground-based instruments. Assimilating these data concurrently within inversion systems will increase our ability to recover net fluxes over smaller regions. However, these instruments must be cross-calibrated against common standards to use these data together, as small inter-calibration differences could potentially strongly impact flux

estimates. We encourage support of these critical cross-calibration activities, as are outlined in Crisp et al. (2018).



## 8.2 Atmospheric CO$_2$ inversions

Atmospheric CO$_2$ inversion analyses are a critical tool for estimating surface fluxes from observations of atmospheric CO$_2$. Expanding observational coverage provides both opportunities and challenges for inversion systems. By addressing the current limitations of our inversion systems, we will be able to take full advantage of increasing observations to improve country-level
top-down estimates of NCE and $\Delta C_{\mathrm{loss}}$. Here we discuss ongoing and planned developments (Sect. 8.2.1), improving model transport (Sect. 8.2.2), missing processes and required assumptions (Sect. 8.2.3), and uncertainty quantification (Sect. 8.2.4).

### 8.2.1 Ongoing and planned developments

To date, there are four operational or quasi-operational atmospheric CO$_2$ inversion systems: CarbonTracker (Jacobson et al., 2020), CAMS (Chevallier et al., 2005b), Jena CarboScope (Rödenbeck et al., 2018) and CMS-Flux (Liu et al., 2021a) that
are regularly updated on annual or quarterly timescales. These systems produce NBE and air-sea flux estimates from either in situ CO$_2$ measurements (CarbonTracker, Jena CarboScope), OCO-2 X$_{\mathrm{CO_2}}$ retrievals (CMS-Flux) or both (CAMS). Similarly, there are seven inversion models (including the aforementioned models) that update CO$_2$ flux estimates annually for the Global Carbon Budget (Friedlingstein et al., 2022), including CAMS (Chevallier et al., 2005b), CarbonTracker Europe (CTE van der Laan-Luijkx et al., 2017), Jena CarboScope (Rödenbeck et al., 2018), UoE in situ (Feng et al., 2016), NISMON-CO2 (Niwa
et al., 2017), MIROC4-ACTM (Saeki and Patra, 2017; Chandra et al., 2021), and CMS-Flux (Liu et al., 2021a).

The OCO-2 MIP activities have semi-regularly performed ensemble inversion experiments (Crowell et al., 2019; Peiro et al., 2022). To date, OCO-2 MIP experiments have been linked to new versions of the ACOS retrieval algorithm, with major improvements to the quality of X$_{\mathrm{CO_2}}$ retrievals occurring during each update. However, as the quality of retrievals have improved (particularly for ACOS v10 onwards), updates to the ACOS retrieval algorithm are becoming less of a driver for new
OCO-2 MIP experiments. In the future, OCO-2 MIP activities could become more regular with annual updates.

The first top-down CO$_2$ system for use in inventory development is CarbonWatch-NZ, under development in New Zealand (https://niwa.co.nz/climate/research-projects/carbon-watch-nz). This program includes expanded CO$_2$ measurement sites and the development of a regional atmospheric CO$_2$ inverse system to quantify the carbon budgets of New Zealand's forest, grassland and urban environments. Initial results suggest stronger uptake by intact forests than estimated through bottom-
up estimates (Steinkamp et al., 2017). This system may serve as an example for other nations through the Integrated Global Greenhouse Gas Information System (IG3IS) framework.

Beyond existing activities, there are a number of planned projects. The European Commission's Copernicus program (https://www.copernicus.eu) has a number of developments ongoing and planned, particularly in building anthropogenic CO$_2$ emissions monitoring and verification support capacity (CO2MVS; Janssens-Maenhout et al., 2020), which is directly linked to
the development and launch of the new CO2M mission and is expected to be operational from 2026 onwards. Further, there are a number of recently completed, ongoing, and planned projects to develop and improve inversion systems to develop operational capacity. Examples include the recently completed CO$_2$ Human Emissions (CHE) project (https://www.che-project.eu/) and follow-up CoCO2 project (https://coco2-project.eu/) that is ongoing, as-well as the VERIFY project (https://verify.lsce.ipsl.fr/).





These projects are developing and refining inversion systems to estimate anthropogenic fossil fuel emissions as well as emissions and removals from the AFOLU sector. Future planned projects include developing approaches to utilize co-emitted species and auxiliary observations ($^{14}$C, solar induced fluorescence, Carbonyl Sulfide, and others) in order to isolate some of the $CO_2$ budget components and improve our understanding of the carbon cycle. For example, multiple data streams could be used together to optimize the dynamic global vegetation model parameters (e.g., Peylin et al., 2016).

In contrast to recent European efforts, there is no mandate for an operational top-down carbon flux-attribution system in the US. Nevertheless, efforts at NOAA centered around CarbonTracker (Jacobson et al., 2020) have been able to produce NBE estimates with relatively low latency harnessing the Agency's substantial flask and in situ $CO_2$ network. In addition, NOAA has developed a higher spatial resolution North American regional inverse system, CarbonTracker-Lagrange (https://gml.noaa.gov/ccgg/carbontracker-lagrange/; Hu et al., 2019). In anticipation of the launch of OCO in 2009, NASA has supported research and development efforts needed to prototype an operational flux estimation system. In particular, the Carbon Monitoring System program (https://carbon.nasa.gov/) has led to the development of both low-latency (2 month) atmospheric $CO_2$ reanalysis (Weir et al., 2021) and approaches to combine top-down NCE estimates with other trace gas measurements (e.g., CO) and non-atmospheric carbon data (e.g., above-ground biomass) to provide improved understanding of carbon cycle processes (Liu et al., 2017; Byrne et al., 2020, 2021; Bloom et al., 2020). There is substantial technical capacity to build an operational system but requires a coordinated effort between federal agencies, academia, and private interests.

In Canada, a prototype operational regional inverse modeling system, the Environment and Climate Change Canada (ECCC) National Carbon Flux Inversion System (ENCIS), is being developed to provide quantitative information on $CO_2$ (and $CH_4$) flux estimates over Canada from national to provincial scales, as well as to understand the carbon cycle in Canada such as $CO_2$ flux in boreal managed and unmanaged forests, wetland emissions of $CH_4$, and GHG emissions over a potentially thawing permafrost in response to the climate change. ENCIS is a regional inverse modeling system based on Lagrangian approach and driven by metrology from the Global Environmental Multiscale (GEM) model (Girard et al., 2014) and is expected to have $1° \times 1°$ spatial resolution.

### 8.2.2 Improving CTM transport

Errors in the representation of atmospheric transport by CTMs has long been recognized as a major source of error in atmosphere $CO_2$ inversion analyses (Law et al., 1996; Law and Simmonds, 1996; Denning et al., 1995, 1999a, b; Baker et al., 2006a; Stephens et al., 2007). Improvements to model transport will provide critical improvements to NCE and $\Delta C_{\text{loss}}$ estimates. Systematic errors in model transport limit our ability to relate surface fluxes and $CO_2$ observations, and can lead to incorrect inferences of surface fluxes (Yu et al., 2018; Schuh et al., 2019; Stanevich et al., 2020). Improving model transport will require work in two areas: (1) improving model parameterizations of unresolved transport, particularly in coarse offline CTMs (like GEOS-Chem run at $4° \times 5°$ in this ensemble) where the spatial and temporal coarsening of meteorological fields can "average-out" vertical transport that is resolved in the parent model (Yu et al., 2018; Stanevich et al., 2020); and (2) increasing spatial and temporal resolution in model simulations, which can better resolve atmospheric transport processes (Agustí-Panareda et al., 2019; Schuh et al., 2019). However, it should be noted that there are limitations to the improvements





that can come from increased model resolution in the global inversion context due to underlying meteorological uncertainties (Liu et al., 2011; Polavarapu et al., 2016, 2018; McNorton et al., 2020). Computational cost is also a significant challenge in inversion systems, because transport models usually scale poorly on supercomputers, for example because of the volume of meteorological data required as input.

As transport models are refined, it will be critical to periodically test their ability to represent large scale atmospheric dynamics. This can be tested using long-lived trace gas species, including sulfur hexafluoride (Schuh et al., 2019), idealized age of air tracer (Krol et al., 2018), and beryllium-7 (Stanevich et al., 2020). Simulations of these trace species are critical in the context of inversion MIPs to gauge inter-model variability and average model bias (Schuh et al., 2019). Similarly, Rn 222 is a useful short lived gas species that enables modelers to evaluate the vertical mixing within the column (Remaud et al., 2018). In addition, model intercomparison studies have proven useful for diagnosing transport errors (e.g., Gaubert et al., 2019; Zhang et al., 2022), and we recommend further activities, such as within the Atmospheric Tracer Transport Model Intercomparison Project (TRANSCOM) framework.

### 8.2.3    Missing processes and required assumptions

The flux estimates provided here do not explicitly account for the atmospheric-chemical production of atmospheric $CO_2$, which occurs from the oxidation of reduced carbon gasses. Instead, these fluxes are either prescribed as surface–atmosphere fluxes (e.g., for FF CO emissions) or neglected from the prior fluxes. This can cause inverse modeling systems to implicitly incorporate the atmospheric $CO_2$ source in optimized surface-atmosphere emissions and removals (i.e. air–sea fluxes and NBE), which can be far from the actual source of the reduced gas. For example, FF CO emissions are largely emitted in the northern extratropics but largely oxidized to $CO_2$ in the tropical troposphere. These incorrectly located emissions of $CO_2$ are large enough to impact top-down inversions (Enting and Mansbridge, 1991; Suntharalingam et al., 2005; Nassar et al., 2010; Wang et al., 2020). Future studies that aim to incorporate an atmospheric source of $CO_2$ would help correct for this current spatial bias (Ciais et al., 2022).

A critical assumption in the top-down $CO_2$ budgets estimated here has been that FF emissions are known and unbiased. Uncertainties in inventory-based FF emission estimates at global and country levels (e.g., Andres et al., 2014) are smaller than top-down NCE estimates; however, inventory-based emission estimates are prone to systematic biases due to the nature of the estimation approach (Guan et al., 2012; Oda et al., 2019) and FF uncertainties could bias the partitioning of NCE between FF and NBE (and propagate into $\Delta C_{loss}$) over countries with large emissions and lower reliability of statistical data collection system, such as China. For example, Saeki and Patra (2017) show that an inferred increase in removals of $CO_2$ by the biosphere over China during 2001–2010 are likely to be an artifact imposed by an error in the trend of anthropogenic $CO_2$ emissions.

### 8.2.4    Uncertainty quantification

The uncertainty in NCE reported here is an estimate of the standard deviation of the v10 OCO-2 MIP ensemble members. This is meant to characterize uncertainties originating from the inversion configuration (such as the transport model, inversion method, and prior constraints). However, there are also limitations to this method. First, there is only a small ensemble





of 11 MIP ensemble members included in this analysis, and an over-representation of inversions using two transport models: TM5 (3) and GEOS-Chem (5), which makes uncertainty quantification challenging. Future approaches that employ "borrowing strength" (Mearns and et al., 2007; Cressie and Kang, 2016) could be employed to better characterize ensemble uncertainty. Second, the ensemble-based uncertainty does not capture some sources of uncertainty. In particular, Bayesian posterior un-

certainties are not considered here (see Sect. 5.2), due to the fact that many of the inversion systems participating in the v10 OCO-2 MIP do not calculate this uncertainty. In addition, we find that the ensemble members that produce Bayesian uncertainties show large differences in magnitude. Thus, this is an area of future improvement for MIP activities, and we recommend more work into characterizing this error component in ensemble inversion experiments. We also note that using an analytic framework, posterior uncertainties and their sensitivities to prior information could be further examined, as has been done for

methane (Worden et al., 2022).

### 8.3 Stockchange estimates

AFOLU emissions and removals are generally quantified as terrestrial carbon stockchanges in managed lands. A number of challenges remain in estimating this quantity from top-down methods. Firstly, lateral fluxes of carbon remain quite uncertain (and associated uncertainty estimates are themselves quite uncertain). The best constrained lateral fluxes are annual country-

level $F_{\mathrm{wood\,trade}}$ and $F_{\mathrm{wood\,trade}}$, which are reported to the UN Food and Agriculture Organization. These fluxes are more uncertain on sub-national scales and sub-annual timescales. Meanwhile, $F_{\mathrm{rivers\,export}}$ is best quantified on basin scales, where stream gauge measurements inform carbon fluxes. Improving sub-national and sub-annual estimates of lateral fluxes would have several benefits: first, this would allow for better sub-national attribution, where regional fluxes could be better quantified. Second, this would allow for incorporating the atmospheric imprint of these carbon fluxes as prior information within

atmospheric $CO_2$ inversion analyses, which may improve flux estimates on sub-national scales.

The Global Stocktake and Paris Agreement do not consider emissions and removals from unmanaged lands. Separating managed lands from unmanaged lands is top-down NCE remains a major challenge, given the smoothed large-scale $CO_2$ flux constraints provided by these top-down methods and the fact that both managed and unmanaged lands can experience considerable stock changes driven by interannual climate variations (e.g., El Niño) and in response to rising $CO_2$ and climate change.

In addition, separating managed and unmanaged lands is further complicated by the fact that there is considerable ambiguity in the definitions managed lands, which can also vary by country (Grassi et al., 2018; Chevallier, 2021). We recommend that each party provide a mask to unambiguously define the plots considered as managed from year to year (Chevallier, 2021).

### 9 Conclusions

We introduced a pilot top-down $CO_2$ budget dataset (Byrne et al., 2022) intended to start a dialogue between research com-

munities and to identify ways that top-down flux estimates can inform country-level carbon budgets. This dataset provides annual country-level and $1° \times 1°$ gridded top-down NCE and $\Delta C_{\mathrm{loss}}$ over 2015–2020, in addition to bottom-up FF and lateral fluxes. These data are provided for four experiments from the v10 OCO-2 MIP that differ in the data used in the assimilation:





IS, LNLG, LNLGIS, and LNLGOGIS. In addition, we provide two metrics for interpreting country-level estimates: (1) the Z-statistic (Sect. 5.1), which quantifies the agreement between IS and LNLG NCE estimates, and (2) the FUR (Sect. 5.2),

which quantifies the impact of atmospheric $CO_2$ data in reducing flux uncertainties.

Country-level flux estimates generally show robust signals for large extratropical countries (e.g., USA, Russia, China). Agreement between the experiments generally decreases for mid-sized countries (e.g., Turkey), particularly in regions with sparse observational coverage for the in situ network (such as the tropics). Large divergences between the IS and LNLG experiments occur in some regions, particularly northern Sub-Saharan Africa, and could be related to the sparsity of in situ

$CO_2$ measurements or biases in OCO-2 retrievals. However, the sparsity of independent $CO_2$ measurements in these regions precludes definitive conclusions. We urge caution in interpreting the $1° \times 1°$ gridded results and suggest collaborating with with experts in atmospheric $CO_2$ inversion systems when using those data.

The accuracy of top-down NCE estimates were characterized through comparisons against independent atmospheric $CO_2$ datasets, and through comparisons against $pCO_2$-based air–sea $CO_2$ fluxes. Overall, the IS, LNLG, and LNLGIS were found

to show the best agreement against independent $CO_2$ measurements, and we recommend using these experiments for analysis. Poorer agreement for experiments assimilating OCO-2 ocean glint $X_{CO_2}$ retrievals, suggesting that residual retrieval biases adversely impact the LNLGOGIS experiment and we urge caution in interpreting these data.

For future GSTs, top-down NCE estimates will be refined as new space-based $X_{CO_2}$ observing systems expand and retrieval algorithms are improved. Complementary expansions of ground-based and aircraft-based $CO_2$ measurements in under-sampled

regions will similarly fill critical observational gaps in regions with large uncertainties and susceptibility to retrieval biases. Improvements to atmospheric $CO_2$ inversion systems, including reductions to systematic transport errors and improved error characterization, will be critical for refining top-down $CO_2$ budgets. And improved estimates of lateral carbon fluxes and managed lands maps will refine estimates of AFOLU emissions and removals.

## 10  Data availability

Top-down $CO_2$ budgets (Byrne et al., 2022) are available from the Committee on Earth Observation Satellites' (CEOS) website: https://doi.org/10.48588/npf6-sw92. Gridded NBE and air-sea fluxes from the OCO-2 MIP are available at https://gml.noaa.gov/ccgg/OCO$_2$_v10mip/. Fossil fuel emissions prescribed in the inversions can be downloaded from https://zenodo.org/record/4776925#.YNX96hNKj2U. The ODIAC2020 emission data product can be downloaded from the Global Environmental Database hosted by the Center for Global Global Environmental Research at NIES

(https://db.cger.nies.go.jp/dataset/ODIAC/DL_odiac2020.html). SeaFlux $pCO_2$-based air–sea fluxes were downloaded from https://zenodo.org/record/5482547#.Yowg18ZlD1I, accessed 23 May 2022.



## Appendix A: TCCON sites

**Table A1.** TCCON sites used for evaluation of posterior $CO_2$ fields of the v10 OCO-2 MIP experiments.

| TCCON site | Country | Latitude | Longitude | Reference |
|---|---|---|---|---|
| Eureka | Canada | 80.05° N | 86.42 °W | Strong et al. (2019) |
| Ny-Ålesund | Norway | 78.9° N | 11.9 °E | Notholt et al. (2019b) |
| Sodankylä | Finland | 67.4° N | 26.6 °E | Kivi et al. (2014) |
| East Trout Lake | Canada | 54.4° N | 105.0 °W | Wunch et al. (2017a) |
| Bremen | Germany | 53.10° N | 8.85 °E | Notholt et al. (2019a) |
| Karlsruhe | Germany | 49.1° N | 8.4 °E | Hase et al. (2014) |
| Paris | France | 48.8° N | 2.4 °E | Te et al. (2014) |
| Orléans | France | 47.9° N | 2.1 °E | Warneke et al. (2019) |
| Garmisch | Germany | 47.5° N | 11.1 °E | Sussmann and Rettinger (2018a) |
| Zugspitze | Germany | 47.3° N | 11.0 °E | Sussmann and Rettinger (2018b) |
| Park Falls | USA | 45.9° N | 90.3 °W | Wennberg et al. (2017) |
| Rikubetsu | Japan | 43.5° N | 143.8 °E | Morino et al. (2014) |
| Lamont | USA | 36.6° N | 97.5 °W | Wennberg et al. (2016b) |
| Anmeyondo | Korea | 36.5° N | 126.3 °E | Goo et al. (2014) |
| Tsukuba | Japan | 36.1° N | 140.1 °E | Morino et al. (2018a) |
| Nicosia | Cyprus | 35.1° N | 33.4 °E | Petri et al. (2020) |
| Edwards | USA | 34.2° N | 118.2 °W | Iraci et al. (2016) |
| JPL | USA | 34.2° N | 118.2 °W | Wennberg et al. (2016a) |
| Caltech | USA | 34.1° N | 118.1 °W | Wennberg et al. (2014) |
| Saga | Japan | 33.2° N | 130.3 °E | Kawakami et al. (2014) |
| Hefei | China | 31.9° N | 117.2 °E | Liu et al. (2018) |
| Izaña | Spain | 28.3° N | 16.5 °W | Blumenstock et al. (2017) |
| Burgos | Philippines | 18.5° N | 120.7 °E | Morino et al. (2018b) |
| Manaus | Brazil | 3.2° N | 60.6 °W | Dubey et al. (2014) |
| Ascension Island | UK | 7.9° S | 14.3 °W | Feist et al. (2014) |
| Darwin | Australia | 12.4° S | 130.9 °E | Griffith et al. (2014a); |
| Réunion island | France | 20.9° S | 55.5 °W | De Mazière et al. (2017) |
| Wollongong | Australia | 34.4° S | 150.9 °E | Griffith et al. (2014b) |
| Lauder 125HR | New Zealand | 45.0° S | 169.7 °E | Sherlock et al. (2014) |





*Author contributions.* The study was conceived of by DC, and designed by BB, DB, SB, KWB, AC, FC, PC, NC, DC, LH, ARJ, JL, JBM, TO, PKP, BP, AS, CS, and JRW. The v10 OCO-2 MIP experiments were performed by DFB, SB, MB, FC, NC, SC, FD, ARJ, RJ, MSJ, DBAJ, JL, ZL, SM, SMM, RM, AS, AZM, and NZ. Lateral flux estimates were performed by FC, PC, ZD, HT, and YY. FF emissions estimates were performed by TO and SB. TCCON data were collected by NMD, MKD, OEG, BH, RK, IM, JN, YSO, HO, CP, KS, KS, YT, VAV, MV, TW and DW. Evaluation of v10 OCO-2 MIP against co-samples was performed by BB, SB, AC, and ARJ. BB wrote the paper and prepared the figures, with contributions from all co-authors.

*Competing interests.* no competing interests are present

*Acknowledgements.* Part of this research was carried out at the Jet Propulsion Laboratory, California Institute of Technology, under a contract with the National Aeronautics and Space Administration (80NM0018D004). The v10 OCO-2 MIP activity was supported by the NASA OCO Science Team program. BB and JL were supported by the NASA OCO2/3 science team program NNH17ZDA001N-OCO2. ARJ, AS, and DFB were funded by NASA award 80NSSC21K1080. AS was also supported by the NASA grant NNX15AG93G. SC acknowledge from the NASA OCO Science Team grant 80NSSC21K1077. The research of NC, AZM, and MB was supported by Australian Research Council Discovery Project DP190100180 and by NASA ROSES grant 20-OCOST20-0004. AZM is also supported by the Australian Research Council Discovery Early Career Research Award DE180100203. Contribution of AC was supported by NASA ROSES Grant numbers 80NSSC20K0006 and 80NSSC21K1068. MSJ acknowledges the internal funding from NASA's Earth Science Research and Analysis Program. The contributions of FC and MR were supported by the Copernicus Atmosphere Monitoring Service, implemented by the European Centre for Medium-Range Weather Forecasts on behalf of the European Commission (grant no. CAMS73), and by the European Union's Horizon 2020 research and innovation programme under grant agreement No 958927 (Prototype system for a Copernicus CO2 service). It was granted access to the HPC resources of TGCC under the allocation A0110102201 made by GENCI. SF at PNNL is supported by the NASA Carbon Monitoring Program (Grant number: 80HQTR21T0069). The Pacific Northwest National Laboratory is operated by Battelle Memorial Institute under contract DE-AC05-76RL01830. NZ acknowledges support from NOAA (NA18OAR4310266) and NASA (80NSSC18K0908). PKP is partly supported by Environment Research and Technology Development Fund (JPMEERF21S20800) of the Environmental Restoration and Conservation Agency of Japan. SMM was supported by the NASA grants 80NSSC18K0976 and 80NSSC21K1073. NMD was funded by ARC Future Fellowship FT180100327

The TCCON Nicosia site has received additional support from the European Union's Horizon 2020 research and innovation programme (grant agreement No. 856612 /EMME-CARE), the Cyprus Government, and the University of Bremen. The TCCON Anmyeondo stie has been funded by the Korea Meteorological Administration Research and Development Program "Developing Technology for Integrated Climate Change Monitoring and Analysis" under grant(KMA2018-00320). TCCON sites at Tsukuba, Rikubetsu and Burgos are supported in part by the GOSAT series project. Burgos is supported in part by the Energy Development Corp. Philippines. The Eureka TCCON measurements were made at the Polar Environment Atmospheric Research Laboratory (PEARL) by the Canadian Network for the Detection of Atmospheric Change (CANDAC), primarily supported by the Natural Sciences and Engineering Research Council of Canada, Environment and Climate Change Canada, and the Canadian Space Agency. The TCCON site at Réunion Island has been operated by the Royal Belgian Institute for Space Aeronomy with financial support since 2014 by the EU project ICOS-Inwire and the ministerial decree for ICOS (FR/35/IC1 to FR/35/C6) and local activities supported by LACy/UMR8105 and by OSU-R/UMS3365 – Université de La Réunion. Darwin



and Wollongong TCCON stations are supported by ARC grants DP160100598, LE0668470, DP140101552, DP110103118 and DP0879468 and NASA grants NAG512247 and NNG05GD07G.

We thank Robert J. Andres for providing uncertainty estimates for CDIAC fossil fuel emission estimates. We thank the data providers of the SeaFlux ensemble for making their pCO2-based air–sea CO2 fluxes publicly available. We are grateful for the leadership of Annemarie Eldering and Mike Gunson of the OCO-2 mission, whose hard work has made this dataset possible.



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
