# Peer review of "National CO2 budgets (2015–2020) inferred from atmospheric CO2 observations in support of the Global Stocktake"

_Earth System Science Data, 2022_

## Author Comment (AC1)

We appreciated the constructive comments of the reviewers. We have addressed the comments below. Reviewer/editor comments are shown in bold with our responses in blue. Line numbers refer to the tracked changes manuscript, and changes to the text are underlined.

Both reviewers asked for the inclusion of a new section demonstrating how the top-down estimates described in this dataset can be compared to National Greenhouse Gas Inventories (NGHGIs) submitted to the UNFCC. Therefore, we have added a new section (Sect. 8) addressing this concern:

L628-657: "8 Comparison with national inventories

[revised manuscript text omitted]

**Reviewer 1**

**General comments**

**Since this study is aimed at "informing countries' carbon budgets", as stated in the abstract, I am missing a bit a section on how the carbon stock changes defined in this study compare to what is required by the UNFCC for national reporting. Moreover, this study calculates the Net Biosphere Exchange (NBE) from the inversions and then adjusts this using the terms for crop and wood trade as well as river export to derive changes in terrestrial carbon stock. The UNFCCC guidelines on reporting also correct carbon stocks for carbon losses due to harvest. For wood, the guidelines include consideration of the turn-over rate of wood products, since many wood products have lifetimes of decades to centuries. For crop, the turn-over time is generally much shorter, order of annual, thus I would think that this term should have little affect on carbon stock changes.**

We have added a section comparing our estimates with NGHGIs (please see above) and explaining that top-down methods do not require an estimate of the turnover rate for harvested wood, as the decomposition of wood products should be implicitly included in the NBE flux. What needs to be accounted for is the lateral movement of wood products from the region where carbon is fixed to the region where the wood products are used and decompose. In contrast, the bottom-up methods described in the UNFCC guidelines need to explicitly estimate the turnover time for wood products, as the release of this carbon to the atmosphere is not captured by these methods. Further, the emissions from wood products are reported by the country that produces the product (under the responsibility of this country). The global sum of wood products used and decompose emissions is thus reported in NGHGIs over the territory of producing countries, but inversions solve for spatial patterns and this is why we included explicitly the displacement of wood products and emissions at the location where they are actually released to the atmosphere. Nevertheless, both approaches should account for the impact of turnover time on wood products in stockchange estimates.

**Specific comments**

**L135: Add a reference for the N-fertilization effect**

Added

**L209: Concerning Eq. 2, why is the carbon in crop harvest considered in the change of carbon stocks, since crop harvest is only a relatively short-term stock of carbon (turn-over on annual time scales) since this will most be consumed or used for fuel?**

When carbon is fixed on land through photosynthesis and then exported laterally and respired elsewhere (e.g., crop growth and harvest, forest growth out of which wood products are harvested), it will result in a local net carbon sink in NBE. However, this does not change the local carbon stocks as this carbon is exported. Therefore, we need to remove this lateral flux (as a global map of sinks where $CO_2$ is fixed and sources where $CO_2$ is emitted after displacement of carbon) of carbon from NBE to calculate the stock change.

**Section 2: How does the carbon stock definition compare to what is required by UNFCCC (see also the general comment)? Also, are the turn-over times of wood products considered in calculating changes in carbon stock?**

Please see new Sect 8 (posted above).

**Eq. 4: May be there is some confusion on my side, but to correct the IQR to standard deviation (SD), I would think one would need to divide by ~1.47, since the IQR includes 50% of the values while one SD includes 34%. Or where does the value of 1.35 come from?**

The conversion from IQR to standard deviation requires the use of normal tables (https://en.wikipedia.org/wiki/Standard_normal_table). We have clarified this in the text, and added a reference to Hoaglin et al. (1985), where a more detailed explanation can be found:

L284-287: "The uncertainty in NCE is calculated as an (denoted $\sigma_{NCE}$) of the distribution's standard deviation using the interquartile range (IQR) of the v10 OCO-2 MIP ensemble. It is a robust estimate that requires only the middle 50% of the ensemble to be normally distributed (Hoaglin et al., 1985). Hence from the normal tables, to two decimal places:"

**L358: What is the reasoning for assuming 30% uncertainty on the crop and wood exchange fluxes?**

This value was chosen based on expert opinion. There has been no rigorous method devised to estimate the uncertainty for these values. We have clarified this in the manuscript:

L385-387: "For each $1° \times 1°$ grid cell, we assume the standard deviation of the mean flux to be 30% for $F_{wood\ trade}$ and $F_{crop\ trade}$, and 60% for $F_{rivers\ export}$. These uncertainty estimates are based on expert opinion as a rigorous error budget has not yet been developed for the $1° \times 1°$ lateral flux estimates."

**L415: Please change "data-model" to "observation-model" if that is what is meant since "data" can be either modelled or observed (also L419). Also, this sentence is a bit ambiguous, do the authors mean that the difference for the OG experiments is more negative in the evaluation data sets compared to experiments?**

We have replaced "data-model" with "observation-model" throughout. We have also clarified the statement about the OG bias:

L438-441: "Over 30°–60° N, where independent observations are densest, we find that the OG ensemble median is biased by -0.69 ppm against TCCON, -0.74 ppm against withheld in situ, and -0.48 ppm against withheld OCO-2 LNLG, suggesting a possible meridional bias (higher retrieved $X_{CO2}$ than independent observations) in the OCO-2 ocean $X_{CO2}$ retrievals."

**Reviewer 2**

**General comments:**

**The authors put together a comprehensive analysis and study on atmospheric CO2 observations, aimed in informing countries' C budgets. The use of atmospheric observations (and OCO-2 MIP outputs in this case) is useful and greatly needed in the view**

**of future CO2M development and in the context of the Paris Agreement global reduction targets. The authors mention often the Paris Agreement and its GST but nowhere in this study UNFCCC reported estimates are presented. Do these atmospheric observations agree or not with the NGHGIs, or better say complement them? How do OCO-2 MIP observations could/will be used for the "informing" purpose? I would highly recommend a section dedicated to this.**

We have added a new section (Sect 8) dedicated to comparing our estimates with NGHGIs. Please see the text for this section posted at the top of this document.

**Specific comments:**

**Introduction: The first three paragraphs contain text widely used in previous studies, references are needed, e.g. VERIFY H2020 project and references therein**

We did not copy any text but likely converged on similar wordings. Nevertheless, we are happy to acknowledge the previous work. We have added references to Petrescu et al. (2021) and Monteil et al. (2020). Please let us know if there are others that we have missed.

Petrescu, A. M. R., McGrath, M. J., Andrew, R. M., Peylin, P., Peters, G. P., Ciais, P., Broquet, G., Tubiello, F. N., Gerbig, C., Pongratz, J., Janssens-Maenhout, G., Grassi, G., Nabuurs, G.-J., Regnier, P., Lauerwald, R., Kuhnert, M., Balkoviˇc, J., Schelhaas, M.-J., Denier van der Gon, H. A. C., Solazzo, E., Qiu, C., Pilli, R., Konovalov, I. B., Houghton, R. A., Günther, D., Perugini, L., Crippa, M., Ganzenmüller, R., Luijkx, I. T., Smith, P., Munassar, S., Thompson, R. L., Conchedda, G., Monteil, G., Scholze, M., Karstens, U., Brockmann, P., and Dolman, A. J.: The consolidated European synthesis of CO2 emissions and removals for the European Union and United Kingdom: 1990–2018, Earth System Science Data, 13, 2363–2406, https://doi.org/10.5194/essd-13-2363-2021, 2021.

Monteil, G., Broquet, G., Scholze, M., Lang, M., Karstens, U., Gerbig, C., Koch, F.-T., Smith, N. E., Thompson, R. L., Luijkx, I. T., White, E., Meesters, A., Ciais, P., Ganesan, A. L., Manning, A., Mischurow, M., Peters, W., Peylin, P., Tarniewicz, J., Rigby, M., Rödenbeck, C., Vermeulen, A., and Walton, E. M.: The regional European atmospheric transport inversion comparison, EUROCOM: first results on European-wide terrestrial carbon fluxes for the period 2006–2015, Atmospheric Chemistry and Physics, 20, 12 063–12 091, https://doi.org/10.5194/acp-20-12063-2020, 2020.

**Line 27: The 5 sectors specified by the IPCC 2006 guidelines are: Energy, IPPU, Agriculture, LULUCF and Waste. The AFOLU (Agriculture + LULUCF) is defined in the 2019 Refinement to the 2006 Guidelines and IPCC ARs reports. However, in their NGHGIs, countries report separates the two sectors.**

AFOLU is defined in the 2006 Guidelines and is the title of volume 4. Nevertheless, we agree that NGHGIs are categorized into Energy, IPPU, Agriculture, LULUCF and Waste. We have revised the text to reflect this:

L23-39: "In support of the first GST, Parties to the Paris Agreement are compiling national greenhouse gas inventories (NGHGIs) of emissions and removals, which are submitted to the United Nation Framework Convention of Climate Change (UNFCCC) and inform their progress toward the emission-reduction targets in their individual NDCs. For these inventories, emissions and removals are generally estimated using "bottom-up" approaches, wherein $CO_2$ emission estimates are based on activity data and emission factors while $CO_2$ removals by sinks are based on inventories of carbon stock changes and models, following the methods specified in the 2006 IPCC Guidelines for National Greenhouse Gas Inventories (IPCC, 2006). This approach allows for explicit characterization of $CO_2$ emissions and removals into five categories: Energy; Industrial Processes and Product Use (IPPU); Agriculture; Land Use, Land-Use Change and Forestry (LULUCF); and Waste. Bottom-up methods can provide precise and accurate country-level emission estimates when the activity data and emission factors are well quantified and understood (Petrescu et al., 2021), such as for the fossil fuel combustion category of the energy sector in many countries. However, these estimates can have considerable uncertainty when the emission processes are challenging to quantify (such as for agriculture, LULUCF, and waste) or if the activity data are inaccurate or missing. For example, Grassi et al. (2022) and McGlynn et al (2022) estimate the uncertainty on the net LULUCF $CO_2$ flux to be roughly 35% for Annex I countries and 50% for non-Annex I countries. In addition, these estimates do not capture carbon emissions and removals from unmanaged systems, which are not directly considered in the Paris Agreement, but impact the global carbon budget and growth rate of atmospheric $CO_2$."

**Line 30: "quantified and understood" please reference**

Added a reference

**Line 40: I would not use "verification system" but complementing, informing..**

We chose this term to be consistent with Chapter 6.10 of the 2019 refinement to the 2006 IPCC guidelines, where atmospheric inversions are discussed.

**Line 60: "several previous studies " how about European BU vs TD studies? Ciais et al., add RECCAP2, as well as Grassi et al., 2022 preprint informing on consistent comparison for the land-se fluxes**

We have added references to Petrescu et al. (2021) and Monteil et al. (2020). The Ciais et al (2021) RECCAP-2 paper was already cited here. We don't believe the Grassi et al. (2022) pre-print is relevant to this paragraph but have added a citation to the first paragraph.

**Line 64: explain CAMS**

Defined acronym

**Line 90: comma (,) before Including**

Done.

**Lines 94-95: The first two sentences don't read well, rephrase please**

We have re-worded these sentences:

L102-108: "The burning of fossil fuels and cement production release geologic carbon to the atmosphere ($40.0 \pm 3.3$ $PgCO_2$ yr$^{-1}$ or $10.9 \pm 0.9$ PgC yr$^{-1}$ over 2010–2019; Canadell et al., 2021). These emissions, along with land use activities, impact carbon cycling between atmospheric, oceanic, and biospheric reservoirs that make up a near-closed system on annual timescales. As a result, roughly half of the emitted $CO_2$ from anthropogenic sources is absorbed by terrestrial ecosystems and oceans…"

**Figure 1 caption: I think number 2 should be moved to forests/logs, to the reservoir itself (under GPP), same as its done for agriculture and water, now its on the urban areas. Add (BB) after biomass burning.**

We agree that this is not an ideal location, but the portion of the figure with the logs has a number of arrows nearby, which makes the number look ambiguous in this location as well. We feel that the current location is better, and the message can still get across with referencing the caption.

**Line 157: can add examples for DGVMs priors**

We have revised DGVM to Terrestrial biosphere model, which is more accurate given that many priors are simple diagnostic models. We have also added some examples:

L165-167: "Prior mean fluxes of net ecosystem exchange are usually obtained from terrestrial biosphere models (such as CASA, ORCHIDEE, and CARDAMOM), while prior mean air-sea fluxes…"

**Line 187 Section 2: a table summarizing all data sources for all lateral fluxes and not only would be of great help.**

We have added a table that lists the data sources and the sub-sections where more details can be found.

**Table 3.** Data sources for lateral flux estimates

| Resolution | Flux | Model / Data source | Section |
|---|---|---|---|
| National | $F_{rivers\ export}$ | Dynamic Land Ecosystem Model (DLEM) and Global NEWS with COSCATs data | Sect. 3.3.1 |
| National | $F_{wood\ trade}$ | UN FAO | Sect. 3.3.2 |
| National | $F_{crop\ trade}$ | UN FAO | Sect. 3.3.2 |
| $1° \times 1°$ | $F_{rivers\ export}$ | Global NEWS with COSCATs data | Sect. 3.3.3 |
| $1° \times 1°$ | $F_{wood\ trade}$ | UN FAO with downscaling | Sect. 3.3.3 |
| $1° \times 1°$ | $F_{crop\ trade}$ | UN FAO with downscaling | Sect. 3.3.3 |

**Line 190: can reference (Fig. 1) after ocean**

Done.

**Line 195: Reco already explained (L116)**

Removed redundant definition

**Line 219: Please mention EU27**

The European Union is meant to be the EU27. We have clarified, "…, the European Union (EU or EU27), …"

**Line 228: add "including 'those' from"...**

Done.

**Line 249: not clear to which network they belong to. In situ collection is referenced as Masarie et al., 2014, perhaps add this ref to the Abstract where you mention first in situ. It appears late in the text (line 185)**

This is a special collection of in situ data made specifically for this activity. They originate from many networks across the globe, we have clarified this in the text:

L297-299: "In situ $CO_2$ measurements (Fig. 3a,d) are drawn from five data collections made available in Obspack format (Masarie et al., 2014). Those source ObsPacks and their references are listed in Table 2. These data include measurements from 55 international laboratories at 460 sites around the world."

**Line 276, Eq 4: what does 1.35 stands for?**

We have clarified this in the text and added a reference to Hoaglin et al. (1985) where a more detailed explanation can be found:

L284-287: "The uncertainty in NCE is calculated as an (denoted $\sigma_{NCE}$) of the distribution's standard deviation using the interquartile range (IQR) of the v10 OCO-2 MIP ensemble. It is a robust estimate that requires only the middle 50% of the ensemble to be normally distributed (Hoaglin et al., 1985). Hence from the normal tables, to two decimal places:"

**Line 322: reference DLEM**

added

**Line 326: please explain DIC, DOC and POC**

Defined: "…that include dissolved inorganic carbon (DIC) of atmospheric origin, dissolved organic carbon (DOC) and particulate organic carbon (POC)."

**Line 358: can you please explain why 30%?**

This value was chosen based on expert opinion. There has been no rigorous method devised to estimate the uncertainty for these values. We have clarified this in the manuscript:

L385-387: "For each 1º ×1º grid cell, we assume the standard deviation of the mean flux to be 30% for $F_{wood\ trade}$ and $F_{crop\ trade}$, and 60% for $F_{rivers\ export}$. These uncertainty estimates are based on expert opinion as a rigorous error budget has not yet been developed for the 1º ×1º lateral flux estimates."

**Line 491: FUR already explained in Fig. 7 caption, then in 5.2, then again here**

Removed this redundant definition.

**Line 538: How about future CO2M mission, what do authors recommend for estimating fluxes at smaller scale (regions, cities etc.)**

The coarse-resolution global models used here are insufficient for monitoring urban emissions from future missions, such as CO2M. These will require higher spatial resolution to resolve urban $CO_2$ plumes from cities. In most cases this will likely require regional models. This is certainly an active area of research, but it is hard to make recommendations based on the analysis here. We can only point out that the methods employed here, such as using an ensemble of inversion systems, will also be useful for quantifying errors in these applications.

**Line 721: perhaps worth mentioning RECCAP2 initiative. I think this paragraph should be in the Introduction**

We have added a mention of RECCAP-2 in this section. We have put it under international activities since it is coordinated through GCP, though with significant funding from ESA-CCI. In addition, we have highlighted the recent co-ordination by WMO:

L813-819: "Finally, there are ongoing internationally organized activities. Phase 2 of the Regional Carbon Cycle Assessment and Processes project (RECCAP-2), coordinated by the Global Carbon Project (https://www.globalcarbonproject.org/reccap/), has aimed to characterize regional carbon budgets. This included investigating how different data sources – including atmospheric inversion analyses – can contribute to this goal (Bastos et al, 2022; Deng et al., 2022). In addition, the WMO has hosted workshops and symposiums with the greenhouse gas monitoring community to develop a framework for sustained, internationally coordinated global greenhouse gas monitoring (e.g., https://community.wmo.int/meetings/wmo-international-greenhouse-gas-monitoring-symposium)."

**Line 807: agree with improving sub-national and sub-annual estimates of lateral fluxes, however its not the aim of the NGHGIs.**

Agreed, this is motivated by improving top-down systems, and this research area may motivate funding of these estimates.

**Line 811: can use 'GST'**

Done.

**Line 817: "We recommend that each party provide a mask" very well thought and optimistic, however very hard to achieve, countries do not invest in it as it's not really required by guidelines, however several newly EU funded projects might look into it for some key countries.**

Agreed that this is hard to achieve, we are happy to hear that EU-funded projects will investigate this.

**Line 824: FUR ok, how about the distribution of the observation network?**

FUR is impacted by the distribution of the observation network, as it is a metric of how much country NCE uncertainties are reduced by the assimilation of available observations. We are unclear on the meaning of the reviewer's question.

**Line 838: a bit more text on how this analysis should inform GSTs and countries' budgets is needed (as for the moment GST is designed for country NGHGIs only), and, as mentioned in the beginning, the UNFCCC and country reported data is missing in this study.**

Please see new Sect 8 (posted at top).

---

## Author Response (AR2)

Dear Dr. Carlson,

**Public justification (visible to the public if the article is accepted and published)**:
Intensely valuable data product and description; thank you for using ESSD.

Thank you, we are glad that this dataset and description are viewed as valuable. We have addressed the minor changes below (with text changes underlined). Our responses are shown in blue.

Please add minor changes as follows:

A few remaining typos. Proofreaders should catch most of these. Suggest that, from group of authors, at least two carefully read proofs to ensure all corrections.

We have checked for typos and will do so again for the proofs.

Missing new section 8 in manuscript summary around lines 80-85?

Added to manuscript summary:

"Section 6 gives a description of the dataset, Sect. 7 shows the characteristics of the dataset, Sect. 8 demonstrates how these data can be compared with national inventories and Sect. 9 discusses current limitations and future directions."

I think you did not define NIES (e.g. line 305 and following)? That particular acronym in this usage might have two meanings?

Added definition:

"…measurements on National Institute for Environmental Studies (NIES) ships…"

Helpful informative data-source landing page, referenced at line 555 and again at line 905. ESSD will want the same link at end of abstract.

Added to the abstract:

These flux and stock change estimates are reported annually (2015–2020) as both a global 1° × 1° gridded dataset and as a country-level dataset and are available for download from the Committee on Earth Observation Satellites' (CEOS) website: https://doi.org/10.48588/npf6-sw92.